# MorphoCellSorter is an Andrews plot-based sorting approach to rank microglia according to their morphological features

Sarah Benkeder[1,2], Son-Michel Dinh[2,3], Paul Marchal[1,2], Priscille De Gea[1,2], Muriel Thoby-Brisson[4], Violaine Hubert[1,5], Ines Hristovska[1,2], Gabriel Pitollat[4], Kassandre Combet[1,2], Laura Cardoit[4], Bruno Pillot[1,5], Christelle Leon[1,5], Marlene Wiart[5,6], Serge Marthy[7], Jérôme Honnorat[1,2,8], Olivier Pascual[1,2]*[†], Jean-Christophe Comte[1,9†]

[1]Claude Bernard Lyon 1 University, Lyon, France; [2]Institut MeLiS, INSERM U1314 CNRS UMR5284, Lyon, France; [3]Institut National des Sciences Appliquées (INSA), Lyon, France; [4]Institut de Neurosciences Cognitives et Intégratives d'Aquitaine, CNRS UMR 5287, Université de Bordeaux, Bordeaux, France; [5]CarMeN Laboratory, INRAE U1397, INSERM U1060, Claude Bernard Lyon 1 University, Lyon, France; [6]CNRS, Lyon, France; [7]Sorbonne Université, Institut du Cerveau – Paris Brain Institute – ICM, Inserm, CNRS, APHP, Hôpital de la Pitié Salpêtrière, Paris, France; [8]French Reference Center on Paraneoplastic Neurological Syndromes and Autoimmune Encephalitis, Hospices Civils de Lyon, Lyon, France; [9]Centre de Recherche en Neurosciences de Lyon (CNRL), INSERM 1028 CNRS UMR5292, Bron, France

*For correspondence:
olivier.pascual@inserm.fr

†These authors contributed equally to this work

## eLife Assessment

The study describes a **useful** tool for assessing microglia morphology in a variety of experimental conditions. The MorphoCellSorter provides a **solid** platform for ranking microglia to reflect their morphology continuum and may offer new insight into changes in morphology associated with injury or disease. While the study provides an alternative approach to existing methods for measuring microglia morphology, the functional significance of the measured morphological changes were not determined.

**Abstract** Microglia exhibit diverse morphologies reflecting environmental conditions, maturity, or functional states. Thus, morphological characterization provides important information to understand microglial roles and functions. Most recent morphological analysis relies on classifying cells based on morphological parameters. However, this classification may lack biological relevance, as microglial morphologies represent a continuum rather than distinct, separate groups, and do not correspond to mathematically defined clusters irrelevant of microglial cells function. Instead, we propose a new open-source tool, MorphoCellSorter, which assesses microglial morphology by automatically computing morphological criteria, using principal component analysis and Andrews plots to score cells. MorphoCellSorter properly ranked cells from various microglia datasets in mice and rats of different ages, from *in vivo*, *in vitro*, and *ex vivo* models, that were acquired using diverse imaging techniques. This approach allowed for the discrimination of cell populations in various pathophysiological conditions. Finally, MorphoCellSorter offers a versatile, easy, and ready-to-use method to evaluate microglial morphological diversity that could easily be generalized to standardize practices across laboratories.

## Introduction

In the healthy brain, microglial cells are characterized by a complex arborization of highly ramified and fine processes that are distributed around the cell body. Their processes present a strong and constant dynamic of protractions and retractions in the adult (*Hristovska et al., 2022*). For many years, this dynamic has been associated with their monitoring role, as they are constantly looking for danger signals in their surroundings (*Davalos et al., 2005*; *Nimmerjahn et al., 2005*). Microglia continuously adapt to their local environment, sensing even small changes via membrane receptors and transporters (*Hanisch and Kettenmann, 2007*). They are able to react quickly and adjust their functions in response to the detected signals. These functional changes are supported by morphological modifications. As an example, the formation of filopodia enables directed migration to lesion sites, while the formation of phagocytic cups and phagolysosomes increases are linked to their phagocytic activity (*Kettenmann et al., 2011*; *Sierra et al., 2013*).

The traditional view of microglia being either ramified (homeostatic microglia) or amoeboid (reactive microglia) has now been proven to be oversimplified and inaccurate, as microglia display a whole range of various morphologies (*Paolicelli et al., 2022*). Indeed, many studies have shown a wide diversity of microglial morphotypes in pathological conditions, and microglia can even exhibit an increase in ramification within the first hours of injury (*Vidal-Itriago et al., 2022*).

In stroke models, morphological transformation of reactive microglia is correlated with the severity and duration of ischemia, as well as an enhanced expression of markers such as CD11B and CD68 (*Zhang, 2019*). In the rodent brain, different morphotypes can be observed depending on how the area has been impacted by the ischemic challenge (*Hubert et al., 2021*). Thus, a gradient of microglial morphologies can be observed from amoeboid microglia in the ischemic core to ramified complex cells in the contralateral cortex (*Anttila et al., 2017*; *Fumagalli et al., 2013*). Similarly, in Alzheimer's disease (AD) mouse models, microglial cells associated with Aβ plaques present strong morphological transformations in opposition to plaque-distant cells mildly morphologically impacted (*Plescher et al., 2018*).

As a matter of fact, morphology is an indicator commonly accessible through live imaging or immunohistochemistry, and it has been used in numerous studies to describe a broad range of pathological contexts (*Adeluyi et al., 2019*; *Ali et al., 2019*; *Morrison et al., 2017*; *Plescher et al., 2018*). To date, the literature contains a wide variety of criteria to quantitatively describe microglial morphology, ranging from descriptive measures such as cell body surface area, perimeter, and process length to indices calculating different parameters such as circularity, roundness, branching index, and clustering (*Adaikkan et al., 2019*; *Heindl et al., 2018*; *Kongsui et al., 2014*; *Morrison et al., 2017*; *Young and Morrison, 2018*). Artificial intelligence (AI) approaches such as machine learning have also been used to categorize morphologies (*Leyh et al., 2021*). This variety of metrics and approaches to measure microglia morphology makes it difficult to compare results between studies. Moreover, categorization methods seem subjective with the number, name, and composition of categories being non-consistent between studies. From a biological and functional perspective, mathematical of manual clustering can be arbitrary as microglia morphology exists along a continuum. In addition, as microglial morphology is highly diverse, it is not rare to be confronted with cells belonging to categories that are not defined, or that could belong to several of them.

Here, we propose a fully automated program that quickly and unbiasedly computes most parameters used in the literature to characterize microglial morphology and overcomes the limitations of current methods. Our approach consists of: (1) performing principal component analyses (PCA) to select the parameters best fitted to describe the dataset of interest and (2) generating a sorting of the cells based on their morphology. The originality of our study is the use of Andrews Plots to rank microglia according to their morphology and discriminate distinct populations. In addition, our working flow has been developed so that it can be used for a wide variety of models and image acquisitions (confocal, wide-field, two-photon, etc.). We developed this method on an exploratory dataset from a rat brain with focal cerebral ischemia and validated it on other independent datasets from other models (other animal, age, pathology) to confirm its wide range of application.

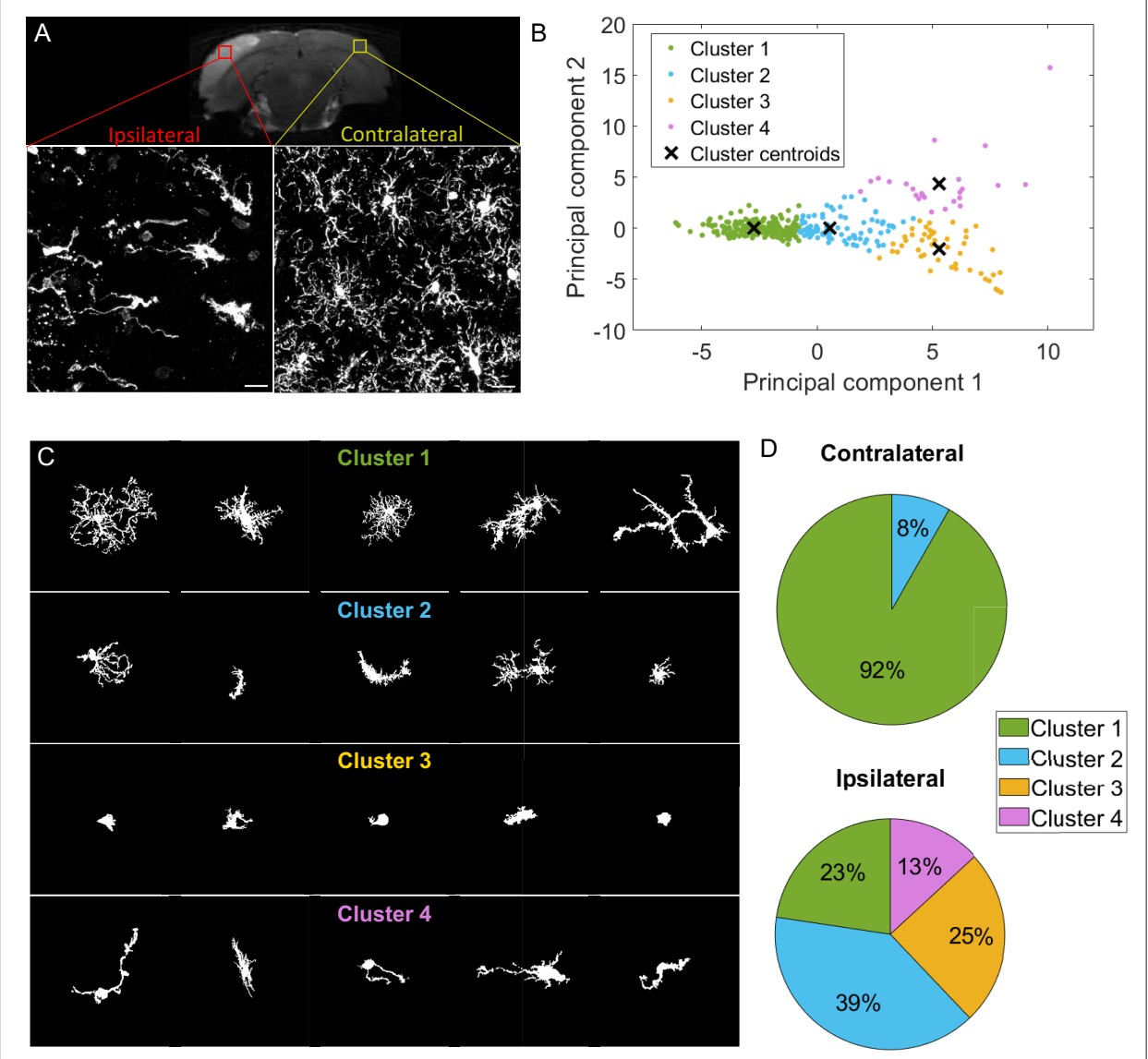

**Figure 1.** k-means clustering identifies highly different populations contralaterally vs ipsilaterally in an ischemic stroke model but offers a heterogeneous cluster composition. (**A**) T2-weighted MRI acquisition of the tMCAo rat brain 24 hr after reperfusion (top). Z maximum projections of IBA1 staining of microglia on the ipsilateral side to the lesion (bottom left) and contralateral side (bottom right). Scale bar: 20 µm. (**B**) Scatter plot with each dot representing one microglia from the complete dataset (ipsilateral and contralateral), plotted in function of their first principal component (PC1) against PC2 values from the principal component analysis (PCA) conducted on all 20 initial morphological parameters. The colors represent the clusters identified by the k-means method, and the 'x' symbols are the centers of each cluster. (**C**) Sample of microglial cells and their affiliated cluster. (**D**) Pie chart illustrating the proportion of each cluster constituting the contralateral and ipsilateral sides.

## Results

### Evaluation of classification methods

Morphological categorization has been at the center of recently developed approaches to characterize microglial morphology. It enables the evaluation of the morphological diversity existing in a cell population. Moreover, in the context of morphological comparison between populations, the study of the morphotypes' proportions is relevant as it can highlight meaningful differences that the sole global comparison of morphological criteria cannot. One way to categorize the cells based on their morphology is to perform clustering to group the cells having similar characteristics. We tested a commonly used approach, k-means, to categorize microglia from a rat brain cortex following an occlusion of the middle cerebral artery (*Figure 1A*). In this model, microglia contralateral to the lesion are

mildly to not impaired, whereas ipsilateral microglia undergo light to drastic morphological changes (*Anttila et al., 2017*; *Fumagalli et al., 2013*; *Figure 1A*). We selected 20 non-dimensional parameters from the literature and developed by the team, on which we conducted a PCA. The k-means clustering algorithm was applied on the first two principal components (PCs), and we fixed the number of clusters at 4, as commonly chosen in the literature (*Verdonk et al., 2016*; *Figure 1B*).

The obtained microglial groups could be named as ramified microglia (cluster 1), with cells having a morphology consistent with a physiological state. Cluster 2 could be referred to as 'reactive microglia,' as it was composed of microglia displaying various morphologies; they had an intermediary ramification level with large cell bodies. Cluster 3 was composed of cells having few short or no processes and corresponded to amoeboid microglia. Lastly, cluster 4 was rod-like microglia, with two processes extending on either side of the cell body or a single long polarized extension. A sample of the cells of the dataset (ipsilateral and contralateral) and their associated cluster is represented in *Figure 1C*. These 4 clusters were not equally represented across each condition, as no amoeboid or rod-like microglia were found in the contralateral side (*Figure 1D*). The large majority of the cells were ramified microglia (92%), and the rest were reactive microglia (8%). The proportion of ramified cells drastically decreased ipsilaterally by a reduction factor of 4 (23%), while reactive microglia increased (39%). 25% of the cells were amoeboid microglia, while rod-like cells represented 13% of the ipsilateral population (*Figure 1D*). These results show that the microglial populations in each region are different, with microglia having a 'non-pathological' morphology mainly found contralaterally, with only a few cells being potentially impacted by the ischemia/reperfusion challenge, as they showed morphological signs of reactivity with a less ramified morphology. Ipsilateral microglia were morphologically highly altered, with a decrease in ramified cells, replaced by reactive and amoeboid microglia, and rod-like microglia.

However, the clusters themselves were highly morphologically heterogeneous, making these quantifications problematic (*Figure 1C*). For instance, Cluster 1 (ramified microglia) was composed of cells with branched processes but also included cells that were visibly affected by the ischemic lesion, exhibiting sometimes short, few, or less branched processes, as well as elongated cell bodies. These microglia might have been more appropriate in Cluster 2 (reactive microglia). The composition of the latter is also questionable, with the presence of branched processes that would fit better in the first cluster, or cells with few very short extensions that resemble some cells found in Cluster 3. Several non-bipolar or non-unipolar cells are also found in Cluster 2. We tested to decrease or increase the number of clusters with no improvement (data not shown).

Other classification methods exist, relying on supervised artificial intelligence (AI) algorithms to identify specific morphotypes. These approaches can have high rates of success (96% for the method developed by Leyh and collaborators *Leyh et al., 2021*), meaning that they are in theory efficiently classifying cells. However, the problems of cells which do not fit into a defined category, or into several categories, can be raised. For example, in our dataset, we found that about 60% of cells would not belong to any category or belong to several categories. In this context, the way the networks are trained might be very subjective to the experimenter and could influence the results. Moreover, depending on the papers, semantic problems are also met, with similar morphologies having different names or, on the contrary same names having different definitions (*Choi et al., 2022*; *Leyh et al., 2021*), making the comparisons between studies complicated.

We know now that microglial morphology is rather a continuum than closed categories (*Paolicelli et al., 2022*). That is why we propose a continuous ranking rather than a classification of the cells.

## Development of a ranking approach based on Andrews plots to order rat fixed microglia in a context of ischemic stroke lesion

We developed MorphoCellSorter, a novel ranking method, on the same dataset of rat microglia in the context of an ischemic stroke lesion, on fixed tissue acquired *via* confocal imaging. This model is known to induce major morphological modifications in microglia (*Hubert et al., 2021*), enabling us to constitute a heterogeneous dataset, with varied morphologies ranging from amoeboid to hyper-ramified.

The method is summed up in *Figure 2A*, and takes as inputs binarized and individualized cells from projected Z stacks. To generate the ranking, we first gathered 20 morphological indices, which we also call parameters, either from the literature (*Clarke et al., 2021*; *Fernández-Arjona et al., 2017*; *Madry et al., 2018*) or developed by the team (see Methods and *Figure 2—figure supplement*

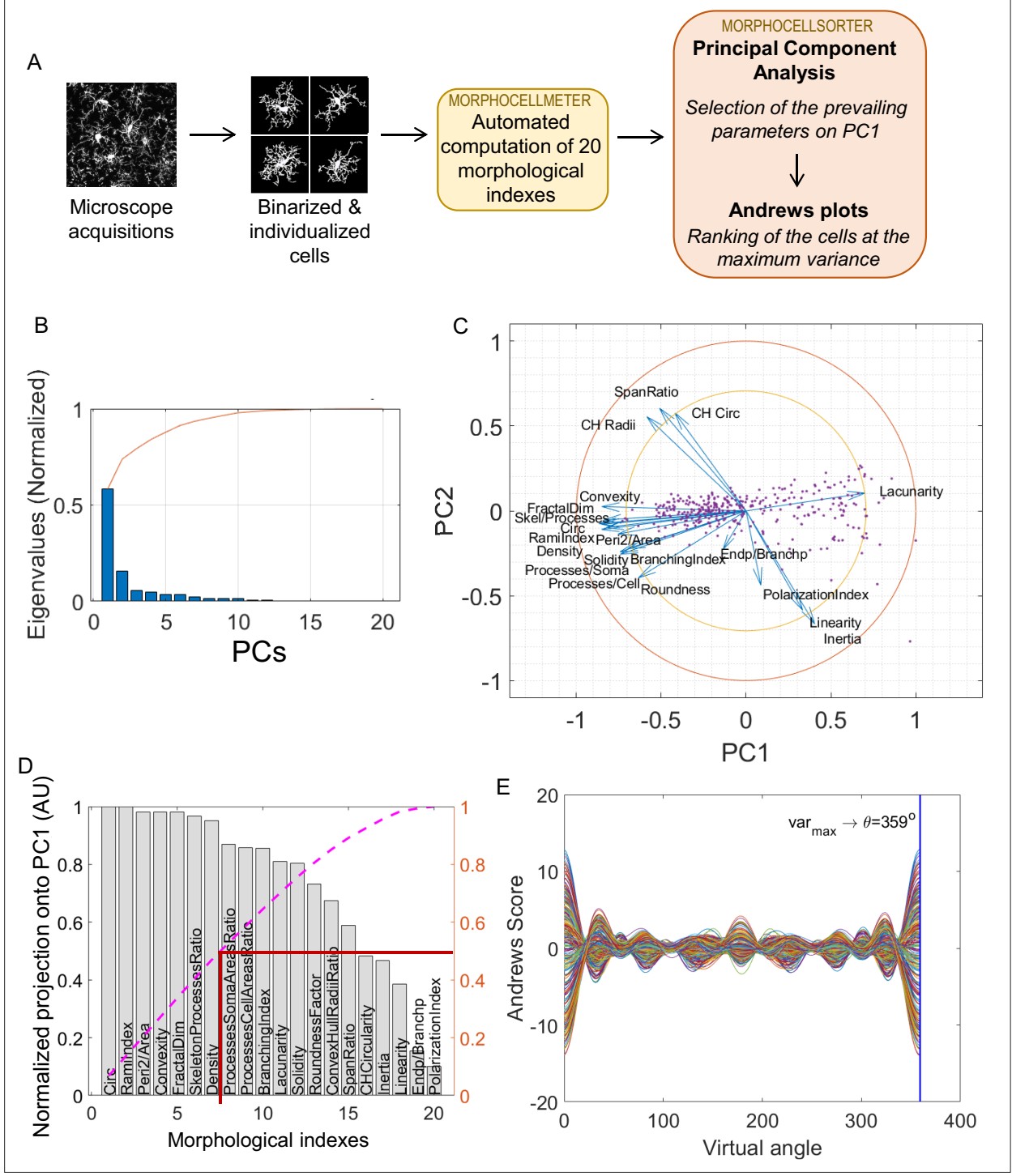

**Figure 2.** MorphoCellSorter approach to generate a ranking of the cells based on their morphology applied to the ischemic stroke model on fixed tissue. (**A**) Summary of the fully automated MorphoCellSorter approach: morphological indexes of the individualized and binarized microglia are computed on a first script (MorphoCellMeter), and the resulting data table is used as an entry for MorphoCellSorter to generate the ranking of the cells based on their morphological characteristics. (**B**) Cascade of normalized eigenvalues. The blue bars represent the eigenvalues for each principal component (PCs), and the red curve is the cumulative sum of the values. The first two eigenvalues corresponding to the first two PCs that account for over 70% of the total sum. (**C**) Correlation circle in the PC1 and PC2 planes. (**D**) Parameters ranked according to their projection on PC1. Values are normalized by the highest projection value. To determine the number of morphological parameters selected, here we consider the median number of the normalized cumulative projection (pink curve), leading to a threshold at 0.5 (red line), and thus seven kept parameters in this case. (**E**) Andrews plots

*Figure 2 continued on next page*

*Figure 2 continued*

generated with the seven parameters weighted according to their contribution onto PC1. Each curve represents one microglia in the high-dimensional space of the seven strongest parameters' projections onto PC1. The maximum variance between curves is obtained at 359°.

The online version of this article includes the following figure supplement(s) for figure 2:

**Figure supplement 1.** Illustration of the 20 morphological indexes automatically computed.

---

*1*). These parameters, describing morphological complexity (circularity, ramification index, roundness factor, perimeter area ratio, density, processes soma area ratio, processes cell area ratio, skeleton processes ratio, branchpoints endpoints ratio, fractal dimension, lacunarity, solidity, convexity, convex hull circularity, convex hull radii ratio, and branching index), linearity (linearity, convex hull span ratio, inertia) and directionality (polarization index) of the cells were normalized to suppress any dimension in order to be exempted from any scale effect. The first step of our method involves PCA, which helps us determine the main parameters responsible for the data dispersion. The first principal component (PC1) captures the direction of maximum variance in the data, and the subsequent component (PC2) captures the direction of the next highest variance (*Figure 2B*). Typically, the first two principal components account for a significant proportion (often over 70%) of the data's total variance, allowing us to reduce the dataset's complexity to a two-dimensional space, such as a two-dimensional plane defined by PC1 and PC2 (*Figure 2C*).

Next, each parameter was projected onto these principal components, which helps us rank parameters based on their influence on the overall data structure (*Figure 2D*). To determine the number of parameters used to elaborate the ranking, we selected the median number of the normalized cumulative projection (threshold = 0.5). We assign weights to the selected parameters based on their projections onto PC1, which allows us to emphasize the most influential parameters. This weighted projection, along with phase factors calculated for each parameter (referred to as $\phi_n$ in the Methods section), forms the basis of an Andrews Plot — a graphical representation that simplifies high-dimensional data into a two-dimensional signal using trigonometric functions.

The Andrews Plot represents each individual microglia, as a unique curve, allowing for visual comparison based on their trajectories within this synthesized signal (*Figure 2E*). This method enables efficient visual comparisons and insights into complex datasets with potentially significant implications for data analysis and interpretation. We took advantage of this representation to rank the cells based on their morphology; the ranking is established at the time point displaying the highest variance, i.e., the time point where the curves are the most dispersed, also named Θ (*Figure 2E*).

The obtained ranking is displayed in *Figure 3A*. To evaluate the quality of the ranking method, we compared the automated ranking to a subjective manual ranking by experts based on visual morphological criteria (cell size; number, length, and branching of the processes). To assess the similarity degree between the rankings, we looked at the distribution of rank differences for each microglia between rankings (*Figure 3B*) and observed that the majority of the cells' rank differences did not exceed an absolute value of 50 rank difference with a peak around 0. We also calculated Spearman's rank correlation coefficient ($r_S = 0.96$; *Figure 3E*). This test revealed that the automated ranking is close to the manual ranking. To ensure the reliability of the manual ranking, we compared the automated ranking with another expert and obtained the same results ($r_S = 0.97$; *Figure 3C and F*). Interestingly, the comparison between the two manual rankings shows similar results to those obtained when compared to the automated ranking ($r_S = 0.93$; *Figure 3D and G*), indicating that variability is the same between automated or manual rankings. The distribution of the rank differences between the same microglial cells in the two rankings elaborated by the experts even shows a greater dispersion of the rank differences, with more cells having rank differences higher than 50 and around 100, showing a part of subjectivity in the cell ranking based on their morphologies. However, automated ranking provided an objective and repeatable result, validating our method. Thus, MorphoCellSorter is able to generate an accurate ranking of the cells based on morphological criteria.

## Evaluation of MorphoCellSorter on other datasets of microglia from different animals, age, models, and imaging techniques

One limitation to the use of a general and common method to characterize microglial morphology is the high diversity of study material: the models (*in vivo*, *ex vivo*, *in vitro*, fixed, or live tissue), the type

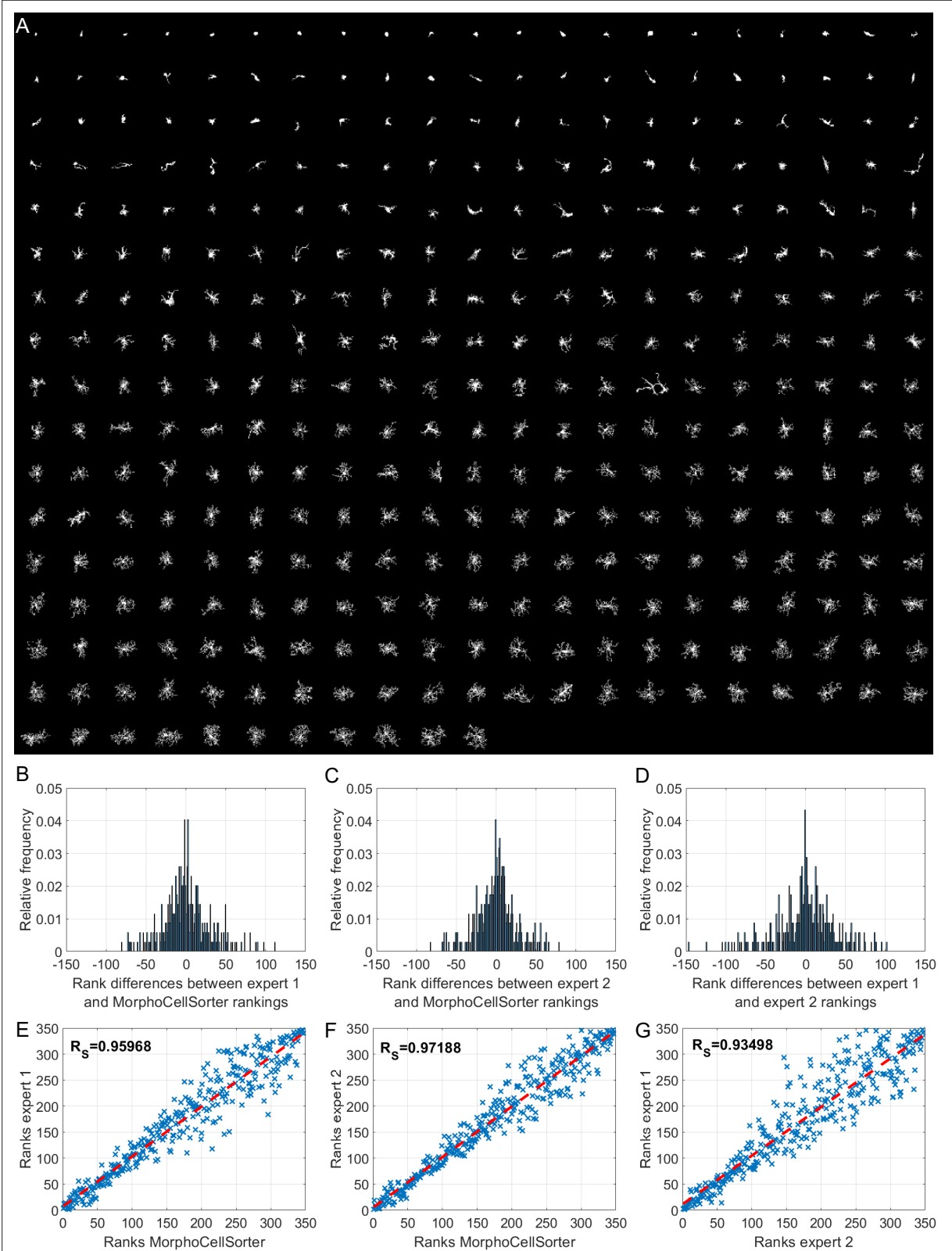

**Figure 3.** MorphoCellSorter offers a reliable ranking of microglia in an ischemic stroke model based on their morphologies. (**A**) Automatic ranking generated by MorphoCellSorter of the 347 microglia constituting the ischemic stroke model dataset. (**B**) Distribution of the rank difference between expert 1 manual and MorphoCellSorter rankings. (**C**) Distribution of the rank difference between expert 2 manual and MorphoCellSorter rankings. (**D**) Distribution of the rank difference between expert 1 and expert 2 manual rankings. (**E**) Correlation between expert 1 and MorphoCellSorter rankings.

*Figure 3 continued on next page*

*Figure 3 continued*

Spearman's correlation coefficient Rs = 95,968, p<0.0001. (**F**) Correlation between expert 2 and MorphoCellSorter rankings. Spearman's correlation coefficient Rs = 97,188, p<0.0001. (**G**) Correlation between expert 1 and expert 2 rankings. Spearman's correlation coefficient Rs = 93,498, p<0.0001.

of collected signal (DAB or fluorescent staining, endogenous fluorescence…), the imaging technique (influencing the resolution and the background noise). We thus wanted to evaluate the capacity of MorphoCellSorter to accurately rank microglia from various datasets having different morphological characteristics (**Table 1**). As previously, we compared the results to manual rankings performed by two independent experts to evaluate the quality of the automated rankings (**Table 1**). We tested the accuracy of the rankings using several thresholds impacting the number of parameters selected as entries for the Andrews plots (**Table 1**). To maximize the accuracy of the automated ranking for all datasets, we determined which threshold provided the best rankings overall (**Table 1—source data 1**). We observed that a threshold of 0.8 (**Table 1—source data 1**), leading to 11–12 selected parameters (**Figure 4A–E**), constituted a good compromise to obtain good rankings for all datasets. Thus, we

**Table 1.** Evaluation of MorphoCellSorter rankings on highly heterogeneous datasets of microglia.
The evaluation of the rankings is conducted by comparing the Spearman's correlation coefficients. w: weeks, m: months, E: embryonic day, P: postnatal day, DIV: day *in vitro*, Ex: expert.

| | | tMCAo fixed tissue | | pMCAo live tissue | | AD model | | Embryonic microglia | | Microglia in culture | |
|---|---|---|---|---|---|---|---|---|---|---|---|
| Model | |  | |  | |  | |  | |  | |
| **Datasets' characteristics** | | | | | | | | | | | |
| Animals | | Male rat 7 w. | | Male mice 8–12 w. | | Female mice 5 m. | | Mice embryos E18.5 | | Mice pups P10 (DIV4-10) | |
| Collected signal | | IBA1 immuno-fluorescent staining | | Endogenous fluorescence CX3CR1GFP/+ | | IBA1 immuno-fluorescent staining | | IBA1 immuno-fluorescent staining | | Brightfield | |
| Imaging technique | | Confocal (40 X) | | Two-photon (20 X) | | Widefield (20 X) | | Confocal (40 X) | | BioStation (40 X) | |
| Fixed/live tissue | | Fixed tissue | | Live tissue | | Fixed tissue | | Fixed tissue | | Live tissue | |
| Number of cells | | n=347 | | n=168 | | n=180 | | n=407 | | n=96 | |
| **Evaluation of MorphoCellSorter rankings** **Correlation with manual rankings** | | | | | | | | | | | |
| Expert | | Ex 1 | Ex 2 | Ex 1 | Ex 2 | Ex 1 | Ex 2 | Ex 1 | Ex 2 | Ex 1 | Ex 2 |
| | 0.2 | 0.965 | 0.958 | 0.945 | 0.945 | 0.952 | 0.934 | 0.914 | 0.885 | 0.854 | 0.749 |
| | 0.3 | 0.964 | 0.954 | 0.944 | 9.946 | 0.953 | 0.942 | 0.914 | 0.885 | 0.854 | 0.749 |
| | 0.4 | 0.964 | 0.964 | 0.935 | 0.938 | 0.949 | 0.947 | 0.929 | 0.909 | 0.845 | 0.744 |
| | 0.5 | 0.96 | 0.972 | 0.938 | 0.939 | 0.943 | 0.945 | 0.927 | 0.907 | 0.877 | 0.837 |
| Thresholds | 0.6 | 0.959 | 0.968 | 0.944 | 0.941 | 0.95 | 0.944 | 0.918 | 0.889 | 0.869 | 0.869 |
| | 0.7 | 0.954 | 0.969 | 0.943 | 0.938 | 0.947 | 0.947 | 0.926 | 0.914 | 0.865 | 0.859 |
| | 0.8 | 0.954 | 0.967 | 0.94 | 0.937 | 0.952 | 0.943 | 0.926 | 0.917 | 0.848 | 0.91 |
| | 0.9 | 0.95 | 0.959 | 0.929 | 0.929 | 0.95 | 0.944 | 0.922 | 0.92 | 0.851 | 0.91 |
| | 1 | 0.95 | 0.959 | 0.922 | 0.923 | 0.954 | 0.946 | 0.913 | 0.923 | 0.846 | 0.906 |
| Ex 1 vs Ex 2 | | 0.935 | | 0.946 | | 0.892 | | 0.912 | | 0.76 | |

The online version of this article includes the following source data for table 1:

**Source data 1.** Data used to determine the optimal threshold.

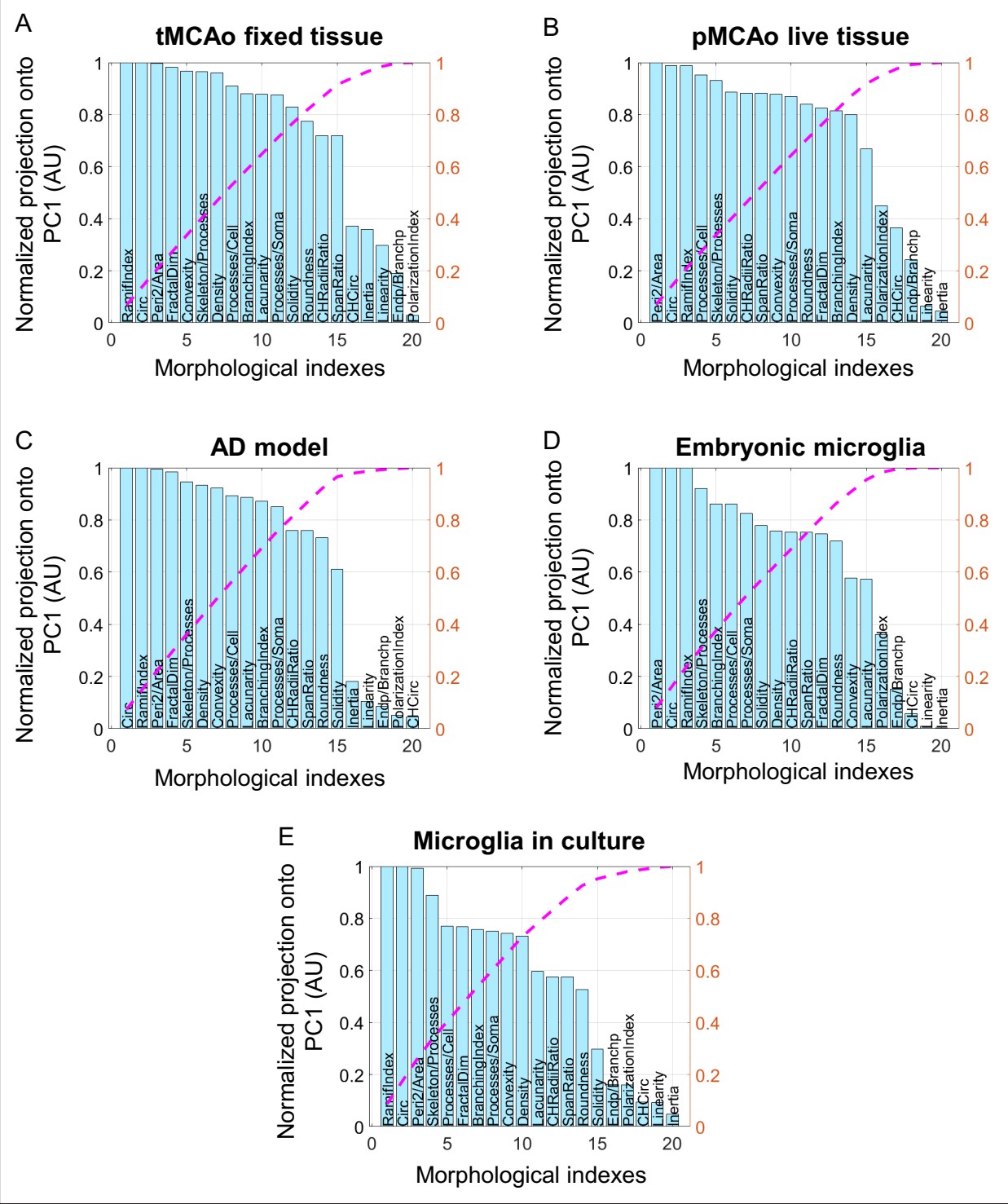

**Figure 4.** Parameters ranked according to their normalized projection onto the first principal component (PC1) for the five tested datasets. Parameters ranked according to their projection onto PC1 for the following datasets: temporary mean cerebral artery occlusion (tMCAo) on fixed tissue (**A**), permanent mean cerebral artery occlusion (pMCAo) on live tissue (**B**), Alzheimer's disease (AD) model (**C**), embryonic microglia (**D**), and microglia in culture (**E**). Values are normalized by the highest projection value. The number of morphological parameters selected is determined by the threshold corresponding value of the normalized cumulative projection selected (pink curve).

The online version of this article includes the following source data and figure supplement(s) for figure 4:

**Source data 1.** Raw data relative to experiments on temporary mean cerebral artery occlusion (tMCAo) model.

*Figure 4 continued on next page*

*Figure 4 continued*

**Source data 2.** Raw data relative to experiments on embryonic microglia.

**Figure supplement 1.** Andrews plots at a 0.8 threshold.

**Figure supplement 2.** MorphoCellSorter rankings generated for temporary mean cerebral artery occlusion (tMCAo) datasets.

**Figure supplement 3.** MorphoCellSorter rankings generated for permanent mean cerebral artery occlusion (pMCAo) and Alzheimer's disease (AD) model datasets.

**Figure supplement 4.** MorphoCellSorter rankings generated for embryonic datasets.

**Figure supplement 5.** MorphoCellSorter rankings generated for microglia in culture.

**Figure supplement 6.** Pre-processing and segmentation procedure for all used datasets.

chose to fix this threshold for the following analyses. The Andrews plots and rankings obtained at the 0.8 threshold are, respectively, displayed in *Figure 4—figure supplements 1–5*.

Thus, our Andrews plot-based ranking method, relying on an automatic selection of relevant parameters thanks to the PCA, allows a reliable automated ranking of the cells based on their morphologies. The diversity of the tested datasets, with different models, types of signal, and imaging techniques shows that our approach offers a good sorting of the cells independently of their shape and context.

## APPxPS1-KI microglia in the visual cortex have significant morphological alterations compared to control cells

To determine the usability of our approach to compare microglial morphology in different conditions, we used MorphoCellSorter to assess the presence of morphological alteration in the context of an adult pathological murine model of Alzheimer's disease. This model is known to induce amyloid plaques with dystrophic neurites, synaptic dysfunction, and behavioral deficits (*Casas et al., 2004*). We compared the morphology of PS1-KI (control) and APPxPS1-KI microglia in the visual cortex (*Figure 5A*). Eleven parameters were automatically selected by the PCA: circularity, ramification index, perimeter area ratio, fractal dimension, skeleton processes ratio, density, convexity, processes cell areas ratio, lacunarity, branching index, and processes soma areas ratio (*Figure 5—figure supplement 1A–C*) and used to generate the Andrews plots (*Figure 5—figure supplement 1D*). The automated ranking obtained revealed that control and APPxPS1-KI microglia were well segregated with APPxPS1-KI microglia at the beginning of the ranking and control mice at the end (*Figure 5B*). The distributions showed a 16.4% overlap between the populations, meaning that a large number of APPxPS1-KI cells have a morphology that differs from that of controls (*Figure 5C*).

The morphological indices showed significant differences between APPxPS1-KI and control microglia. The parameters measuring the morphological complexity of the cells highlighted a clear decrease in the ramification level of APPxPS1-KI microglia, with, for example, a fractal dimension and convexity decrease and a circularity increase. However, there were no significant differences in the cells' linearity and polarity between groups (*Figure 5D–W*).

Thus, this approach allowed the detection of morphological changes of microglia in the visual cortex of APPxPS1-KI mice.

## MorphoCellSorter pinpoints no morphological alterations in a model of congenital central hypoventilation syndrome

Congenital central hypoventilation syndrome (CCHS) is a rare genetic disease associated with mutations of the PHOX2B gene and characterized by life-threatening breathing deficiencies (*Ceccherini et al., 2022*; *Dubreuil et al., 2008*). Phox2b is a transcription factor required for the specification of the autonomous nervous system that contains respiratory control centers. Because some of our preliminary observations strongly suggest a possible involvement of microglia in the disease (unpublished data), we tested whether MorphoCellSorter highlighted eventual morphological alterations of microglia in Phox2b mutants (CCHS embryos) compared to wildtype embryos (WT), focusing on brainstem tissue that hosts respiratory networks involved in breathing rhythmogenesis.

In this case, using a threshold of 0.8, the PCA highlighted 12 parameters that were used to rank microglia with Andrews plots: circularity, ramification index, perimeter area ratio, skeleton processes ratio, fractal dimension, processes cell areas ratio, density, branching index, processes cell areas ratio,

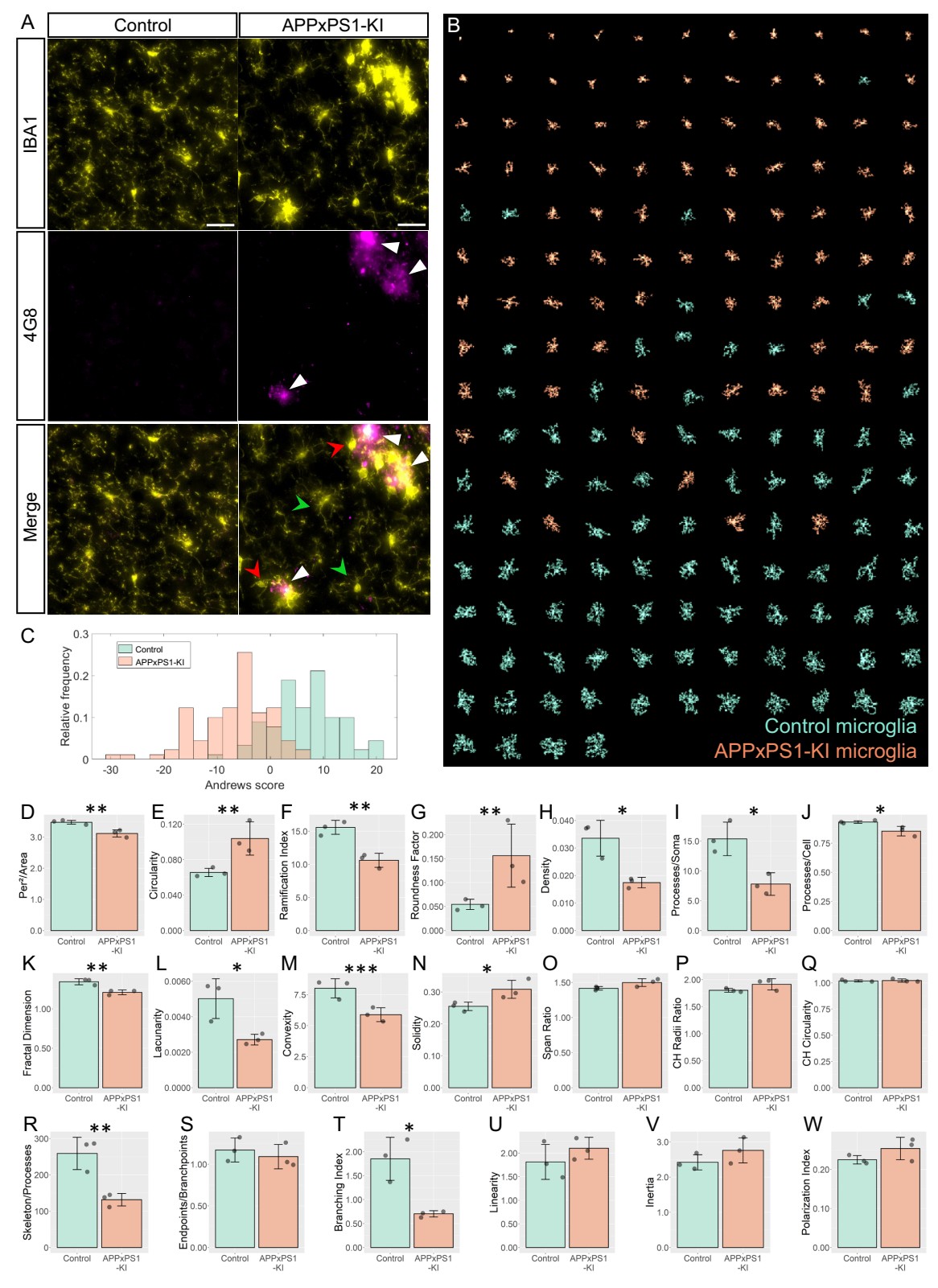

**Figure 5.** MorphoCellSorter identifies morphological alterations in the visual cortex of APPxPS1-KI mice. (**A**) Z maximum projections of IBA1 (microglia) and 4G8 (Aβ plaques) stainings of control and APPxPS1-KI mice in the visual cortex. The white arrowheads point at Aβ plaques positive for 4G8, green arrowheads designate ramified microglia far from Aβ plaques, and red arrowheads point to highly morphologically altered microglia close to Aβ plaques. Scale bar: 20 μm. (**B**) MorphoCellSorter automated ranking. Control microglia are represented in green and APPxPS1-KI microglia in orange.

*Figure 5 continued on next page*

*Figure 5 continued*

(**C**) Distribution of the Andrews values at the maximum variance (Andrews scores) for control and APPxPS1-KI microglia. (**D–W**) Morphological indexes computed by MorphoCellSorter. The data shown are the mean ± standard deviation (std). Linear mixed models were applied when the application conditions were respected and Wilcoxon tests were performed otherwise (for the Span Ratio) (N=3 mice for each condition, $n_{APP-PS1xKI}$ and $n_{Control}$ = 180). *p<0.05; **p<0.01; ***p<0.001.

The online version of this article includes the following source data and figure supplement(s) for figure 5:

**Source data 1.** MorphocellSorter Data relative to experiments on the Alzheimer's disease (AD) model.

**Source data 2.** Data relative to the statistical analysis of the experiments related to the Alzheimer's disease (AD) model.

**Figure supplement 1.** Elaboration of MorphoCellSorter ranking for the Alzheimer's disease (AD) dataset.

convexity, lacunarity, and solidity (*Figure 6—figure supplement 1A–C*). These parameters were weighted to generate the Andrews plots (*Figure 6—figure supplement 1D*) used to rank the cells according to their morphology (*Figure 6B*; complete ranking in *Figure 6—figure supplement 2* A-B).

Both populations displayed a large variability of morphologies from amoeboid to a few ramified processes and were well represented throughout the ranking (*Figure 6B*). This is concordant with previous studies on developing microglia: they can concomitantly be found in a large variety of shapes and morphologies (*Orłowski et al., 2003*). Indeed, the distributions of Andrews scores showed no difference between WT and mutant mice. The overlap of distributions between the populations found was 78.9% (*Figure 6C*). Accordingly, when taking into consideration the various morphological indices, there was no significant difference between WT and Phox2b mutant microglial morphologies (*Figure 6D–W*). Additional experiments are required to test whether this is specific for embryonic stages and whether this is associated or not with different microglial functional states.

## Discussion

MorphCellSorter enables the automated measurement of morphological descriptors, the calculation of several non-dimensional indices, and a meaningful comparison of cell populations based on their morphological characteristics. The method was initially applied to an ischemic stroke model on confocal images with significant morphological changes. MorphoCellSorter was subsequently tested on five different datasets with heterogeneous characteristics (various models, age, sex, imaging technique, and type of collected signal), showcasing its adaptability. We then successfully used our approach to compare the cells' morphology in an Alzheimer's disease context and in embryonic microglia using a CCHS model. Our method can autonomously identify the most effective parameters for discriminating the studied cell set, thereby providing valuable insights into microglial morphology. Additionally, it offers a useful automated ordering of cells based on a combination of selected parameters using Andrews plots. This feature allows for precise positioning of cells along an axis, ranging from round cells to highly branched cells, facilitating the discrimination of various cell populations based on their morphologies.

The use of the Andrews plots offers the possibility to combine all automatically selected most discriminant parameters and to weight them according to their relevance, making the rankings more accurate and less subjective than only sorting the cells based on one parameter. Even if we recommend a threshold at 0.8 for the number of parameters selected, as it gave good rankings for all datasets we tested, we included the possibility to change this threshold if the generated ranking is not satisfying, knowing that all thresholds give accurate rankings when compared to expert rankings.

The automated parameter selection relies on principal component analysis, which is used to determine the parameters that best disperse the data. In contrast, other approaches use PCA for dimensionality reduction to create new morphological parameters that are difficult to apprehend as they do not refer to any physical measure (*Clarke et al., 2021*; *Heindl et al., 2018*; *Salamanca et al., 2019*). The strength of MorphoCellSorter lies in its personalized approach, with automated parameter selection adapted to the dataset, making it easily applicable to segmented microglia in diverse contexts, including those with distinct morphologies, such as microglia in culture or *in vivo* acquisitions. It does not require program adaptation or the need to train a neural network for deep/machine learning approaches. We observed that some selected parameters were consistent between all the tested datasets (ramification index and perimeter area ratio, for example), showing their robustness to

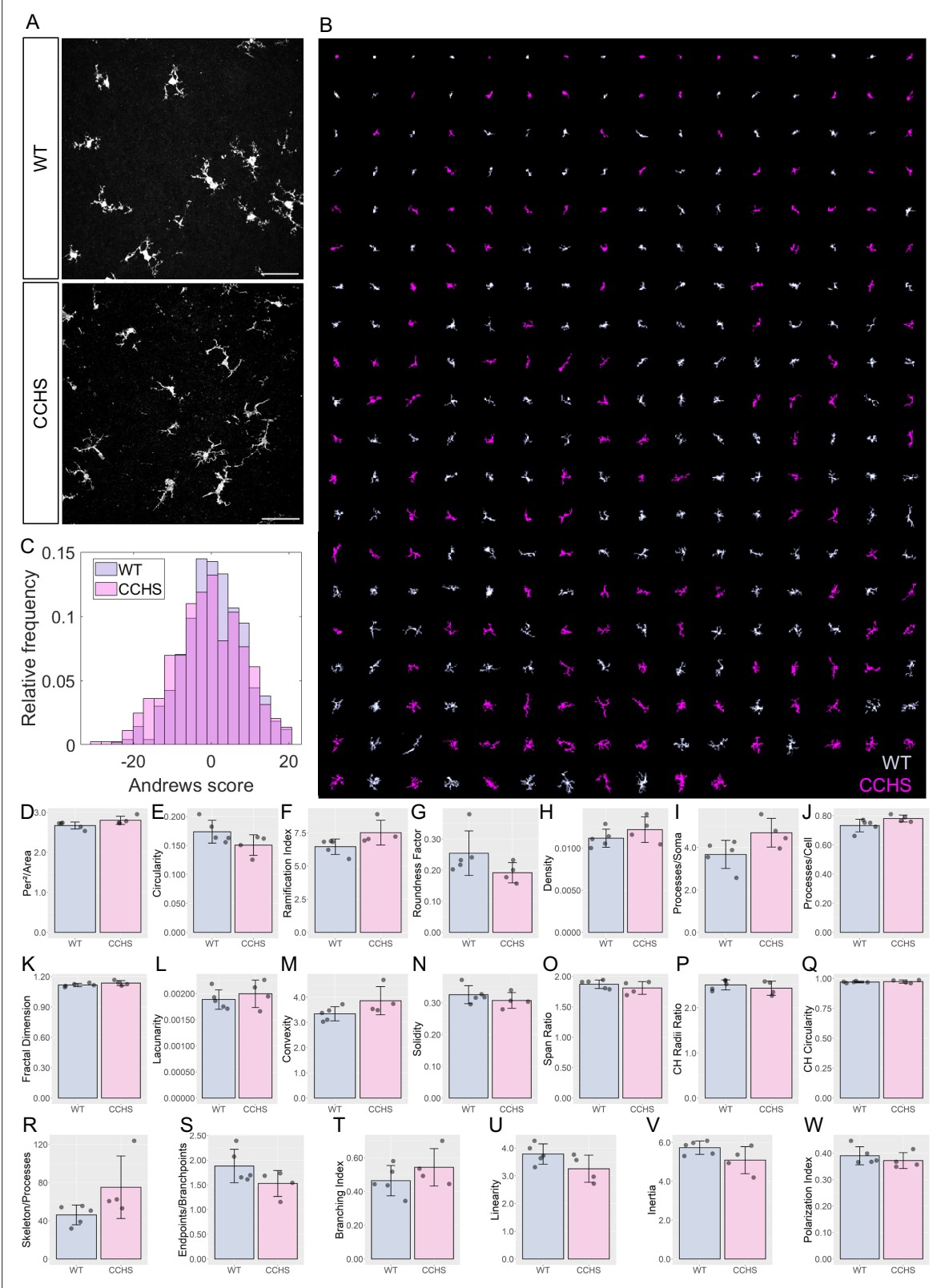

**Figure 6.** Accurate sorting of embryonic microglia reveals no morphological difference between central congenital hypoventilation syndrome (CCHS) and wild-type (WT) embryos' microglia. (**A**) IBA1 stainings in WT and CCHS embryos' brainstems. (**B**) MorphoCellSorter automated ranking. WT microglia are displayed in light purple and mutated microglia are represented in pink. Only every 3 microglia composing the dataset are displayed for readability purposes. (**C**) Distribution of the Andrews scores for CCHS and WT microglia. (**D–W**) Morphological indexes computed by MorphoCellSorter.

*Figure 6 continued on next page*

*Figure 6 continued*

The data's distribution is represented as well as the mean ± standard deviation (std). Linear mixed models were applied ($N_{WT}$ = 5, $N_{mutant}$ = 4; $n_{WT}$ = 498 and $n_{mutant}$ = 444).

The online version of this article includes the following source data and figure supplement(s) for figure 6:

**Source data 1.** MorphoCellSorter data relative to experiments on embryonic microglia.

**Source data 2.** Data relative to the statistical analysis of the experiments related to embryonic microglia.

**Figure supplement 1.** Elaboration of MorphoCellSorter ranking for the central congenital hypoventilation syndrome (CCHS) dataset.

**Figure supplement 2.** MorphoCellSorter automated ranking of the central congenital hypoventilation syndrome (CCHS) dataset.

differentiate microglial cells with various morphologies. On the contrary, some parameters were never selected (inertia and linearity, for example). The information contained in these parameters were not discriminant enough in the datasets we used in this present work, but we can imagine that they would have been selected if we would have chosen a model in which microglial growth towards a specific site is observed, such as focal laser injury (*Davalos et al., 2005*; *Haynes et al., 2006*).

We also introduce a ranking tool to further enhance morphological analysis. In contrast to recent studies, we opt for ordering cells rather than classifying them. Microglial morphologies can span a continuum with no clear boundaries between morphotypes. Automated clustering methods such as k-means (*Young and Morrison, 2018*) may be more suggestive, as they require an *a priori* determination of the number of clusters. Hierarchical clustering (*Clarke et al., 2021*) can avoid this requirement but may result in non-reproducible clusters, or clusters without biological meaning. Attempts to identify distinct morphotypes, as previously done with artificial intelligence (*Leyh et al., 2021*), are also questionable due to varying definitions of morphotypes between laboratories and a lack of consideration for subtle morphological differences and intermediate forms. This problem is overcome by the ranking approach, as it does not lock the cells in defined categories but rather position the cells in relation to one another. However, the ability to compare the evolution of clusters of microglial morphologies in different conditions remains valuable, especially when morphologies are heterogeneous in the same population, such as in developmental microglia or pathological contexts. Comparing medians or means of certain parameters may not reveal differences due to high variability, whereas comparing the evolution of the proportions of specific morphotypes may provide more informative insights. The Andrews plots used to generate a ranking of the cells could also be used to generate clusters of the cells presenting morphological resemblances, as similar cells display similar Andrews curves' trajectories.

We chose not to integrate a segmentation tool into our program, as the segmentation method should be tailored to the specific imaging conditions, materials, and microscopes used. Diverse segmentation methods are available in different software, with varying levels of automation and manual input requirements. Although automated thresholding would be ideal. In our case, image acquisitions were not entirely uniform, even within the same sample. For instance, in ischemic brain samples, lipofuscin from cell death introduces background noise that can artificially impact threshold levels. This effect is observed even when comparing contralateral and ipsilateral sides of the same brain. In our experience, manually adjusting the threshold provides a more accurate, reliable, and comparable selection of cellular elements, even though it introduces some subjectivity. To ensure consistency in segmentation, we recommend that the same person performs the analysis across all conditions. In this work, we present methods using FIJI. It is important to note that achieving perfect cell segmentation is not a prerequisite for using MorphoCellSorter since the tool does not aim to provide highly precise measurements of microglial morphology. For instance, discontinuous processes are not problematic. Consequently, it is possible to work with lower-resolution microscopes, such as epifluorescence wide-field microscopes, although the segmentation process may be more labor-intensive if the source images are of lower quality. Obviously, the results of the ranking will be highly affected by the quality and the homogeneity of segmentation in a given data set. However, we did not assess how much the resolution of the microscope used could affect the quality of the ranking as low resolution and poor signal-to-noise ratio would mainly affect the segmentation step which is not the purpose of this paper.

One potential limitation is that we decided to work with 2D Z-stack projections, as accurate 3D reconstructions are time-consuming and require high-resolution acquisitions. Our goal was to develop

a fast and robust method that could be broadly applicable to most microscopes and imaging scenarios. Therefore, our approach may not yield precise measurements of parameters such as the exact number and length of processes or their hierarchy. However, other research groups have successfully developed tools capable of providing these types of measurements, such as 3DMorph (*York et al., 2018*) and MorphOMICS via Imaris (*Colombo et al., 2022*), Mic-Mac (*Salamanca et al., 2019*). Instead, MorphoCellsorter is a rapid way to determine if a treatment, genotype or location of cells affects morphology without going through the laborious work of 3D reconstruction. Our tool can thus be complementary to methods that provide those precise metrics.

In summary, MorphoCellSorter is an efficient and user-friendly tool for morphological analysis that operates with minimal computation time. The code is open-source on GitHub, accessible by the community (https://github.com/Pascuallab/MorphCellSorter, copy archived at *Marchal and Benkeder, 2025*) and improved according to the needs. It can be complemented with additional measures, such as the expression of specific markers, to investigate the relationship between morphology and gene expression. The emergence of spatial transcriptomics is a promising avenue for furthering our understanding of the link between morphology and phenotype in the microglia field in which MorphoCellSorter will be of great interest.

## Methods
### Models
All experimental procedures were carried out in accordance with the French institutional guidelines and ethical committee, and authorized by the local Ethics Committees of the University of Bordeaux and 'Comité d'éthique pour l'Expérimentation Animale Neurosciences Lyon:' CELYNE; CNREEA no. 42, and the Ministry of National Education and Research (APAFIS#30666–2021032518063894).

### Ischemic stroke rat model – fixed tissue
The experiment was conducted on 6–8 wk-old male Sprague Dawley rats with transient (90 min) middle cerebral artery occlusion (tMCAo) followed by reperfusion. Twenty-four hours after reperfusion, the animal was deeply anesthetized (xylazine at 12 mg/kg and ketamine at 90 mg/kg in 0.9% NaCl) and died by the collapse of the lungs when the thoracic cage was open. Infusion of phosphate-buffered saline (PBS) was performed directly in the left ventricle. Next, 100 mL of paraformaldehyde (PFA) 4% was infused at 10 mL/min before the brain collection. The brain was then post-fixed in PFA 4% for 2–4 d, rinsed, and soaked in 30% sucrose for 72 hr before being frozen in –40 °C 2-methylbutane. It was then stored at –20 °C until being cut. Floating 30 µm thick sections were obtained using an NX50 cryostat and conserved in PB-Thimerosal until immunostaining.

### Ischemic stroke mouse model – live tissue
An ischemic stroke was induced in 8–12 wk Cx3cr1$^{+/GFP}$ male mice (RRID:IMSR_JAX:005582) by a permanent occlusion of their middle cerebral artery (pMCAo). Imaging was performed the next day directly on living animals through a cranial window. Briefly, animals were anesthetized with isoflurane (induction: 3–4%, surgery: 1.5–2%) and installed on a stereotaxic apparatus where their temperature was monitored and maintained at 37 °C. After cleaning and exposition of the skull, a polyamide implant was glued on a zone at the periphery of the lesion in order to be able to image both ischemic and extralesional regions. The skull was then cautiously thinned by drilling until a thickness of 20–30 µm was reached. A glass coverslip was then glued to the thinned region. Mice were then put under a heating lamp and individually housed for 24 hr until imaging (*Hubert et al., 2021*).

### Phox2b mutation and embryonic dataset
The expansion of a 20-residue polyA tract in PHOX2B is used as a model for Congenital Central Hypoventilation Syndrome (CCHS) (*Amiel et al., 2003*). Mutant pups were generated by mating conditional Phox2b$^{27Ala/27Ala}$ females with Pgk::Cre males (*Dubreuil et al., 2008*). Since mutant newborns die rapidly after birth, all experiments were performed blind on E18.5 embryos of either sex, and the genotype of each embryo was determined *a posteriori* on tail DNA. Pregnant mice were killed by cervical dislocation at E18.5. Embryos were rapidly excised from uterine horns and placed in artificial oxygenated cerebrospinal fluid (aCSF) at 18–20°C until dissection. The aCSF solution composition

was (in mM): 120 NaCl, 8 KCl, 0.58 NaH2PO4, 1.15 MgCl2, 1.26 CaCl2, 21 NaHCO3, 30 Glucose, pH 7.4, and equilibrated with 95% $O_2$-5% $CO_2$. Embryos were decerebrated and decapitated, and ventral tissues were removed. Brainstems were then carefully disassociated from the surrounding tissues and completely isolated by a rostral section made at the junction between the rhombencephalon and mesencephalon and a caudal section at the spinal cord level. After dissection, brainstem preparations were placed in 4% PFA for 2–3 hr for tissue fixation. To obtain transverse frozen sections, brainstems were placed in a 20% sucrose-PBS solution for cryoprotection overnight. Then they were embedded in a block of Tissue Tek (Leica Microsystems, France) and sectioned at 30 µm using a cryostat (Leica Microsystems, France).

## Alzheimer's disease model

C57BL/6 5-mo-old female APPxPS1-KI and control (PS1-KI) mice were used (*Casas et al., 2004*). For the brain collection, 0.5 mg xylazine was injected intraperitoneally, followed by Euthasol. They were then intracardially perfused first with 4% PFA in PB. After 1 hr, the brains were collected and then post-fixed for 2 hr in 4% PFA and stored in PBS. They were rinsed in phosphate buffer (PB) and soaked in 30% sucrose for 48 hr before being frozen at –40 °C in 2-methylbutane for 2 min and stored at –70 °C until the cut. Free-floating sections (30 µm thick) were cut using a cryostat and conserved in PB-Thimerosal until immunostaining.

## *In vitro* microglia

We followed the protocol published by Bohen and collaborators using Magnetic Activated Cell Sorting (MACS) (*Bohlen et al., 2019*). Ten-days-old C57BL6/J pups were anesthetized with isoflurane, intracardially perfused with PBS enriched with magnesium and calcium (14040141, Gibco) before decapitation and brain collection. The brains were then finely chopped in a dissociation buffer and transferred to a tissue grinder to further dissociate the cells from the tissue. Myelin and debris were then eliminated thanks to a Percoll PLUS solution (E0414, Sigma-Aldrich) diluted with DPBS10X (14200075, Gibco) and enriched in $MgCl_2$ and $CaCl_2$ (for 50 mL of myelin separation buffer: 90 mL of Percoll PLUS, 10 mL of DPBS10X, 90 µL of 1 M $CaCl_2$ solution, and 50 µL of 1 M $MgCl_2$ solution). An incubation with anti-myelin magnetic beads (130-096-433, Miltenyi Biotec): the cell suspension is applied on a column fixed to a magnet so that the myelin will stay on the column while the cells will exit. To isolate specifically microglia, the suspension was incubated with anti-CD11b magnetic beads (130-097-142, Miltenyi Biotec). 20.000 cells were then seeded on each of the four compartments of the 35 mm diameter dishes with a polymer bottom (80416, Ibidi) coated with poly-D-lysine (100 µg/mL, A-003-E, Sigma-Aldrich) and collagen IV (2 µg/mL, 354233, Corning). The culture medium was adapted from *Bohlen et al., 2019* with DMEM/F12 (21041025, Gibco), 1% penicillin-streptomycin, 2 mM L-Glutamine (10378016, Gibco), 5 µg/mL N-acetylcysteine (A9165, Sigma-Aldrich), 100 µg/mL apo-transferrin (T1147, Sigma-Aldrich), 100 ng/mL sodium selenite (S5261, Sigma-Aldrich), 1.5 µg/mL cholesterol (700000 P, Avanti), 0.1 µg/mL oleic acid (90260, Cayman Chemical), 1 ng/mL gondoic acid (20606, Cayman Chemical), 1 µg/ml heparan sulfate (AMS.GAG-HS01, AMSBIO), 2 ng/mL TGFβ2 (100-35B, Preprotech), and 10 ng/mL M-CSF (315–02, Preprotech). Cells were kept at 37 °C, 5% $CO_2$, and the culture medium was half renewed every 2 d until imaging (4–10 d after seeding).

## Immunohistochemistry

The sections underwent 5 min rinses three times with PBS and Triton 100X 0.3% (PBS-T). Sections from the ischemic experiment were incubated for 20 min in a 1.5% hydrogen peroxide solution diluted in PBS, and after 3*10 min of rinsing with PBS-T, they were incubated for 30 min in 0.5% sodium borohydride diluted in PBS and then thoroughly rinsed. One-hour saturation was then performed on all sections with PBS-T, 3% bovine serum albumin, and 5% normal goat serum (PBS-T-BSA-NGS). Sections were then incubated overnight with anti-Iba1 antibodies (1/500, Wako, Germany, #W1W019-19741) at 4 °C and anti-Amyloid-β 4G8 (1/500, Bio-Legend, #800709) antibody. Sections were rinsed 3*10 min the next day with PBS-T. They were then incubated in goat anti-rabbit secondary antibodies coupled with Alexa 555 fluorochrome (1/300, Invitrogen, France, #A21429) and goat anti-mouse coupled with Alexa 488 fluorochrome (1/300, Invitrogen, France, #A11029) for 2 hr at room temperature. They were then rinsed 3*10 min with PBS and mounted on microscope slides using Fluoromount G (Thermo Fisher Scientific, #00-4958-02) and a glass coverslip.

For the Phox2b mutant model and embryonic dataset, sections were incubated for 90 min in a solution of PBS containing 0.3% Triton X-100 and 1% BSA to limit nonspecific labeling and favor antibody tissue penetration. The primary rabbit anti-Iba1 antibody (1/500; Wako, Germany, #W1W019-19741) was then applied overnight at room temperature and under slight agitation. To amplify the primary antibody signal, after several washes, sections were incubated with an Alexa Fluor 568 donkey anti-rabbit secondary antibody (1/500, Merck, France, #A10042) for 90 min at room temperature. Immunostained sections were then mounted in Vectashield Hard Set medium (Eurobio, France), coverslipped, and kept in the dark until imaging.

## Image acquisitions

For the ischemic stroke model on fixed tissue, acquisitions were performed with a Zeiss confocal microscope and a 40 X objective (LSM 880, Zeiss). Z-stacks between 18 and 25 µm thickness were executed with a 1 µm step. For the ischemic stroke model on live, microglia were observed thanks to a two-photon microscope (Bruker Ultima) and a 20 X objective, at a depth of 50–150 µm. 10–15 µm thick acquisitions were acquired with a 1 µm step between plans. For the AD model, a Zeiss Axio Scan 7 microscope slide scanner was used with a 20 X objective. Z-stacks (30 µm thick) were performed with a step of 1 µm. For the Phox2b mutant model, imaging was conducted using a Zeiss confocal microscope (LSM 900, Zeiss). Z-stacks (30 µm thick) were performed with a step of 1 µm and a 40 X objective. *In vitro* microglia were imaged on a single plane using the BioStation (Nikon).

## Image processing and segmentation

The general procedure is described in *Figure 4—figure supplement 6*.

For the ischemic model on fixed tissue, all post-acquisition processing was executed with FIJI software (2.9.0/1.54 f). Individual microglial cells were cropped from the original acquisitions before Z-maximum projections. The brightness of the projections was adjusted, and median (radius = 2), mean (radius = 2), and unsharp mask (radius = 5; mask = 0.60) filters were applied via a macro. A threshold was then applied to each crop to binarize the cells, and pixels not belonging to the microglia of interest were manually deleted. The same procedure was performed for the Phox2b mutant model, except the mean filter that was not applied.

For the ischemic model on live tissue, all post-acquisition processing was also executed with FIJI software. Z-maximum projections were performed, and individual crops of the cells were done. Brightness and contrast of the images were modified before applying a manual threshold to binarize the cells. Reminiscent noise and neighboring cells were manually removed.

For the AD model, crops of individual microglial cells located in the secondary visual cortex were extracted from images using the *Zen* software (v3.5; Zeiss) and exported to the Tif image format. The next operations were executed with FIJI software. Z stack cleaning was performed by background subtraction (rolling = 50), followed by a median filter (radius = 3) and an unsharp mask (radius = 5; mask = 0.60). Segmentation was performed by the FIJI Trainable Weka Segmentation 3D plugin (*Arganda-Carreras et al., 2017*). The selected training features were based on the Gaussian blur, Laplacian, Maximum, Mean, Minimum, Median, and Variance (Minimum sigma: 1.0 and Maximum sigma: 8.0). Binarized z-stacks were then manually cleaned to remove pixels from neighboring cells or remaining noise. Finally, a maximum Z-projection was performed.

For the Phox2b mutant model, image processing was performed via a custom MATLAB R2018B (Mathworks, Massachusetts, USA) script. First, a median filter followed by an unsharp mask filter was applied to all images. An automated threshold was then applied. Manual tuning with a thresholding tool was applied if the automated threshold was not satisfying. The binarized images were then automatically cropped around microglia. Briefly, the centroid of each detected object (i.e. microglia), except the ones on the borders, were detected, and a crop of 300×300 pixels around the objects were generated. Then, the pixels belonging to neighboring cells were manually removed on each generated crop.

For the *in vitro* microglia, acquisitions were cropped around individual microglia, and the cells were then manually traced on Procreate (5.2.9 version).

When the binarization process generated one or several holes inside the cell bodies, they were manually filled to not interfere with the automated detection of the cell bodies. All binarized and individualized microglial cells were then rescaled through a FIJI macro command to obtain a pixel

resolution equal to 0.5 µm. Finally, the images were resized to obtain a field size equal to 300×300 pixels.

## MorphoCellSorter algorithm pipeline

### Automated measure of 20 dimensionless morphological descriptors

All parameters are illustrated in *Figure 2—figure supplement 1*.

First, we calculate the morphological parameters most commonly used in the literature, followed by other parameters that we have developed and proposed here (*Figure 2B*). To make our approach scale-independent, we reformulated some morphological descriptors to obtain dimensionless parameters.

$$Circularity = \frac{2 \times \sqrt{\pi \times Cell\,Area}}{Cell\,Perimeter}$$

$$Ramification\,Index = \frac{\frac{Cell\,Perimeter}{Cell\,Area}}{2 \times \sqrt{\pi \times Cell\,Area} \times Image\,length \times Image\,width}$$

$$Roundness\,Factor = \frac{4 \times Soma\,Area}{\left(Longest\,axis\,length\right)^2 \times \pi}$$

$$Perimeter\,Area\,Ratio = \frac{Perimeter^2}{Cell\,Area}$$

$$Density = \frac{Cell\,Area}{Length \times width\,of\,the\,image}$$

$$Inertia = \frac{Longest\,axis\,length}{Smallest\,axis\,length}$$

### Cell body recognition

The cell body identification is based on successive erosions until the center of the cell body is the only pixels remaining using a 3*3 pixels square as a structuring element. The last erosion is the one preceding the elimination of all pixels belonging to the cell and corresponds to the center of the cell body. Then, successive dilations are performed to reconstruct the cell body according to the morphology of the cells, using a cross (+) 3 pixels height and width (*Leyh et al., 2021*). Reconstruction of the cell body relies on (1) the identification of the index *i* on which measured surfaces after erosions are inferior to the surfaces measured after dilation, (2) the average of two copies of the images: one on which *i*+1 erosions have been performed, and the other one on which *i* dilatations have been performed from the center of the cell body. The resulting mean image was then subjected to a threshold fixed at 0.25 to obtain the cell body. From this, we can calculate the following parameters.

$$Processes\,Soma\,Areas\,Ratio = \frac{Processes\,Area}{Soma\,Area}$$

$$Processes\,Cell\,Areas\,Ratio = \frac{Processes\,Area}{Cell\,Area}$$

$$Polarization\,Index = \frac{Soma\,Centroid\,|\,Cell\,Centroid}{\sqrt{Cell\,Area}}$$

### Skeleton

The cell skeleton is obtained via the MATLAB *bwskel* function. This function reduces all 2-D binary objects to 1-pixel width curves while taking care of not changing the essential structure of the image. The aim is to extract the image skeleton while keeping the topology properties such as the Euler number and to extract branchpoints and endpoints. The skeleton length is defined by the number of pixels forming the skeleton minus the pixels of the cell body.

Thus, we calculate the skeleton-related parameters:

$$Skeleton\,Process\,Ratio = \frac{\left(Skeleton\,Length\right)^2}{Processes\,Area}$$

$$Branchpoints\,Endpoints\,Ratio = \frac{Number\,of\,Branchpoints}{Number\,of\,Endpoints}$$

## Sholl analysis

Sholl analysis was performed on the microglial skeleton. Concentric circles with a step size of two pixels were generated on the skeletons minus the soma, and the soma centroid was the center point of the circles. Intersections between the growing circles and the skeleton were counted.

$$Branching\,Index = \sum \frac{(Intersections\,circle_n - Intersections\,circle_{n-1}) \times r_n}{\sqrt{Length \times width\,of\,the\,image}}$$

## Fractal/Hausdorff dimension

The fractal dimension measures the complexity of a cell shape, that is, how an object fills the space at different scales. The Hausdorff fractal dimension (**Costa, 2023**) corresponds to the slope of the logarithm of the number of tiles of size ε needed to cover all the surfaces of the object of interest vs the logarithm of ε.

## Lacunarity

Lacunarity corresponds to the inhomogeneity of a cell shape. It measures the variance of how the object fills the space at different scales. Patterns with larger gaps generally have higher lacunarity. We used the gliding box algorithm (**Vadakkan, 2023**; **Tolle et al., 2008**), and the value corresponds to the slope of the log-log representation.

## Convex hull

The convex hull is the smallest convex polygon (with angles inferior to 180°) enclosing the whole cell.

$$Solidity = \frac{Cell\,Area}{Convex\,hull\,Area}$$

$$Convexity = \frac{Cell\,Perimeter}{Convex\,hull\,Perimeter}$$

$$Convex\,Hull\,Circularity = \frac{2 \times \sqrt{\pi \times Convex\,hull\,Area}}{Convex\,hull\,Perimeter}$$

We also measure the length of the major and minor axes of the convex hull.

$$Convex\,Hull\,Span\,Ratio = \frac{Length\,Major\,axis}{Length\,Minus\,axis}$$

Finally, we measure the largest and smallest radii of the circles from the centroid of the convex hull as the center to external points.

$$Convex\,Hull\,Radii\,Ratio = \frac{Maximum\,Convex\,hull\,radius}{Minimum\,Convex\,hull\,radius}$$

## Linearity

A principal component analysis is generated on all the points forming the microglial cell. Then, the variance of the two first PCs is calculated; the highest corresponds to the Variance Max and the lowest to the Variance Min.

$$Linearity = \frac{Variance\,Max}{Variance\,Min}$$

## Establishment of the cell ranking according to morphological criteria

The second step is to determine which of the previously calculated parameters best discriminate the cells in a given dataset. PCA is a statistical method that allows dataset dimensionality reduction (**Jolliffe and Cadima, 2016**). A PCA is performed on the 20 morphological indexes calculated to determine the parameters that best explain the dispersion of the data. Lacunarity, roundness factor,

convex hull radii ratio, processes cell areas ratio, and skeleton processes ratio were subjected to an inversion operation in order to homogenize the parameters before conducting the PCA: Indeed, some parameters rank cells from the least to the most ramified, while others rank them in the opposite order. By inverting certain parameters, we can standardize the ranking direction across all parameters, thus simplifying data interpretation. To identify the main parameters responsible for the data spreading, we perform a PCA.

The principal component 1 ($PC_1$) is the direction in which data is most dispersed. The two first components represent most of the information (about 70%), hence we can consider the plan $PC_1$, $PC_2$ as the principal plan reducing the dataset to a two-dimensional space. Next, we calculate the projection of the parameters $F_n$ onto the two first principal components, respectively, $PC_1$ and $PC2$ (**Figure 2C**) and rank those parameters according to the intensity (absolute value) of their projection onto $PC_1$ (**Figure 2D**). Each parameter is then weighted by a factor $W_n = w_n./w_0$, where $w_n$ is the absolute value of the $n^{th}$ parameter projection onto $PC_1$, and $w_0$ is the absolute value of the strongest projection among all the parameters. As shown in **Figure 2C**, each parameter $F_n$ is represented by a vector in the $PC_1$, $PC_2$. Thus, we calculate the projection $y_n$ of each parameter onto $PC_2$ and calculate the following phase factor.

$$\phi_n = \arctan(Y_n/W_n).$$

The following stage of our method consists in introducing the previously calculated factors (Weight: $W_n$, and phase: $\phi_n$) and their corresponding parameter into a special Fourier synthesis called 'Andrews Plot' or 'Andrews curve' (**Andrews, 1972**; **García-Osorio and Fyfe, 2005**) allowing us to visualize the data structure in high-dimensional. The general form of the spectral Fourier synthesis of an individual $j$ writes:

$$S_j(t) = \sum_{n=1}^{N} A_n \cos(2\,\pi n\,t) + B_n \sin(2\,\pi\,n\,t),$$

that we rewrite as,

$$S_j(t) = \sum_{n=1}^{N} C_n \cos(2\,\pi n\,t + \phi_n),$$

where $n$ is the index of the ranked parameters, $N$ is the full number of parameters $F_n$, $t$ is a virtual time $0 \leq t \leq 1$, and $C_n = F_n \times {}_{Wn}$. Finally, we derive this method to represent data in high dimensional by a simple one-dimension signal composed of orthogonal trigonometric functions in researching the time $t=t^*$ for which the curves present the maximum of variance. At this specific time point, we rank individuals according to the magnitude of their own function $S_j(t^*)$ value as represented in **Figure 2E**. We now call $S(t^*)$ the 'Andrews-Score.'

## Statistics

Rankings comparisons, performed with Spearman's correlation, were also used to measure the monotonic correlation between the rankings. They were executed with MATLAB.

We used mixed-effects models on RStudio, that are adapted to complex experimental designs with dependent and non-dependent data. This allows the addition of random factors to fixed effects. Model fittings and analysis were performed with the lme4 package (**Amiel et al., 2003**). Condition (Control/APP-PS1xKI or WT/CCHS) was considered a fixed effect, and the factor mice was considered a random intercept. When the mixed models were not applicable, we used Wilcoxon tests. Statistical significance is shown on the graphs (*p<0.05; **p<0.01; ***p<0.001).

Number of bins of Andrews scores histograms was automatically determined by the Freedman-Diaconis method based on the following formula:

$$k = (x_{max} - x_{min})/(2 \times IQR \times N^{(-1/3)})$$

With $x_{max}$ and $x_{min}$ being, respectively, the maximum and minimum Andrews scores, IQR the inter-quartile range, and N the number of cells.

## Availability of data and materials

The original code of MorphoCellSorter is available on GitHub (https://github.com/Pascuallab/MorphoCellSorter, *Marchal and Benkeder, 2025*).

## Acknowledgements

We acknowledge the contribution of SFR Santé Lyon-Est (UAR3453 CNRS, US7 Inserm, UCBL) facility: CIQLE (a LyMIC member), especially Bruno Chapuis and Denis Ressnikoff for their help. This study is supported by research grants from Fondation pour la recherche médicale (reference DQ20170336751) and has been developed within the BETPSY project, which is supported by a public grant overseen by the French national research agency (Agence nationale de la recherche, ANR), as part of the second 'Investissements d´Avenir' program (reference ANR-18-RHUS-0012). This work was also supported by the French National Research Agency within the framework of the LABEX CORTEX ANR-11-LABX-0042.

## Additional information

### Competing interests

Muriel Thoby-Brisson: Reviewing editor, eLife. The other authors declare that no competing interests exist.

### Funding

| Funder | Grant reference number | Author |
|---|---|---|
| Fondation pour la Recherche Médicale | DQ20170336751 | Jérôme Honnorat Olivier Pascual |
| Agence Nationale de la Recherche | ANR-18-RHUS-0012 | Jérôme Honnorat |
| Agence Nationale de la Recherche | ANR-11-LABX-0042 | Olivier Pascual |

The funders had no role in study design, data collection and interpretation, or the decision to submit the work for publication.

### Author contributions

Sarah Benkeder, Data curation, Formal analysis, Writing – original draft, Writing – review and editing; Son-Michel Dinh, Data curation, Formal analysis, Methodology; Paul Marchal, Data curation, Formal analysis, Writing – review and editing; Priscille De Gea, Formal analysis; Muriel Thoby-Brisson, Resources, Data curation, Validation, Writing – review and editing; Violaine Hubert, Ines Hristovska, Laura Cardoit, Bruno Pillot, Christelle Leon, Resources; Gabriel Pitollat, Resources, Data curation; Kassandre Combet, Data curation, Formal analysis; Marlene Wiart, Serge Marthy, Resources, Validation, Writing – review and editing; Jérôme Honnorat, Funding acquisition, Writing – review and editing; Olivier Pascual, Conceptualization, Resources, Data curation, Software, Formal analysis, Supervision, Funding acquisition, Validation, Methodology, Writing – original draft, Project administration, Writing – review and editing; Jean-Christophe Comte, Conceptualization, Data curation, Software, Formal analysis, Supervision, Validation, Investigation, Methodology, Writing – original draft, Writing – review and editing

### Author ORCIDs

Paul Marchal ⓘ https://orcid.org/0000-0003-2821-8119
Muriel Thoby-Brisson ⓘ https://orcid.org/0000-0003-3214-1724
Violaine Hubert ⓘ https://orcid.org/0000-0001-5656-5690
Marlene Wiart ⓘ https://orcid.org/0000-0003-0326-7002

Olivier Pascual https://orcid.org/0000-0002-5730-9880

### Ethics

All experimental procedures were carried out in accordance with the French institutional guidelines and ethical committee, and authorized by the local Ethics Committees of the University of Bordeaux and "Comité d'éthique pour l'Expérimentation Animale Neurosciences Lyonn": CELYNE; CNREEA no. 42, and the Ministry of National Education and Research (APAFIS#30666-2021032518063894).

Reviewer #1 (Public review): https://doi.org/10.7554/eLife.101630.3.sa1
Reviewer #2 (Public review): https://doi.org/10.7554/eLife.101630.3.sa2
Author response https://doi.org/10.7554/eLife.101630.3.sa3

---

## Additional files

### Supplementary files
MDAR checklist

### Data availability

All data generated or analysed during this study are included in the manuscript and supporting files; source data files have been provided for figures.

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
