## [Editor Report · eLife Assessment]

The study describes a **useful** tool for assessing microglia morphology in a variety of experimental conditions. The MorphoCellSorter provides a **solid** platform for ranking microglia to reflect their morphology continuum and may offer new insight into changes in morphology associated with injury or disease. While the study provides an alternative approach to existing methods for measuring microglia morphology, the functional significance of the measured morphological changes were not determined.

---

## [Referee Report · Reviewer #1 (Public review)]

The current manuscript by Bendeker et al. (2024) presents a new platform, MorphoCellSorter, for performing population wide microglial morphological analyses. This method adds to the many programs/platforms available to determine characteristics of microglial morphology; however, MorphoCellSorter is unique in that it uses Andrew's plotting to rank populations of cells together (in control and experimental groups) and present "big picture" views of how entire populations of microglia alter under different conditions. In their ranking system, Bendeker et al. (2024) use PCA to determine which of the morphological characteristics most define microglial populations, avoiding user subjective biases to determine these parameters. Compared to "expert" evaluators, MorphoCellSorter appears to perform consistently and accurately, including in different types of tissue preservation methods and in live cells, a key feature of the program. In addition, the researchers point out that this platform can be used across a wide array of imaging techniques and most microscopes that are available in a basic research lab. There are minor concerns about the platform's utility in analyzing embryonic microglia and primary microglial cultures, but overall, this platform will be another useful tool for microglial researchers to consider using in future studies. Furthermore, the method of morphological assessment aligns with the current direction of the field in identifying microglial cells in more nuanced ways.

In their current revision, the authors have done an excellent job responding to concerns and have updated the manuscript accordingly.

---

## [Referee Report · Reviewer #2 (Public review)]

The authors introduce MorphCellSorter, an open-source tool available on GitHub, designed for automated morphometric analysis of microglia. Current understanding suggests that microglia represent a heterogeneous population, especially in non-steady adult states, better characterized as a continuum rather than distinct cell groups.

This tool was developed to classify microglia along this continuum. Using stained brain sections and microscope imaging, individual microglia are binarized and processed with MorphCellSorter, which categorizes them based on 20 morphological parameters. Notably, the tool is versatile, as it can be applied to both fluorescent and brightfield brain sections, as demonstrated by the authors. Additionally, it has been tested across various setups (both fixed and live tissues) and biological contexts (including embryonic stages, Alzheimer's disease models, stroke, and primary cell cultures), showcasing its versatility and adaptability. Overall, the study is well-conceived and could have some value in the field.

Numerous similar tools already exist, and the number is likely to grow, especially with advancements in AI. These tools have limited scientific utility as they provide descriptive rather than informative outputs. Microglial morphology varies due to external influences (such as developmental stages and injuries), but the significance of these variations remains largely hypothetical.

---

## [Author Response]

The following is the authors’ response to the original reviews.

**Public Reviews (consolidated):**
In the microglia research community, it is accepted that microglia change their shape both gradually and acutely along a continuum that is influenced by external factors both in their microenvironments and in circulation. Ideally, a given morphological state reflects a functional state that provides insight into a microglia's role in physiological and pathological conditions. The current manuscript introduces MorphoCellSorter, an open-source tool designed for automated morphometric analysis of microglia. This method adds to the many programs and platforms available to assess the characteristics of microglial morphology; however, MorphoCellSorter is unique in that it uses Andrew's plotting to rank populations of cells together (in control and experimental groups) and presents "big picture" views of how entire populations of microglia alter under different conditions. Notably, MorphoCellSorter is versatile, as it can be used across a wide array of imaging techniques and equipment. For example, the authors use MorphoCellSorter on images of fixed and live tissues representing different biological contexts such as embryonic stages, Alzheimer's disease models, stroke, and primary cell cultures.This manuscript outlines a strategy for efficiently ranking microglia beyond the classical homeostatic vs. active morphological states. The outcome offers only a minor improvement over the already available strategies that have the same challenge: how to interpret the ranking functionally.

We would like to thank the reviewers for their careful reading and constructive comments and questions. While MorphoCellSorter currently does not rank cells functionally based on their morphology, its broad range of application, ease of use and capacity to handle large datasets provide a solid foundation. Combined with advances in single-cell transcriptomics, MorphoCellSorter could potentially enable the future prediction of cell functions based on morphology.

Strengths and Weaknesses:(1) The authors offer an alternative perspective on microglia morphology, exploring the option to rank microglia instead of categorizing them with means of clusterings like k-means, which should better reflect the concept of a microglia morphology continuum. They demonstrate that these ranked representations of morphology can be illustrated using histograms across the entire population, allowing the identification of potential shifts between experimental groups. Although the idea of using Andrews curves is innovative, the distance between ranked morphologies is challenging to measure, raising the question of whether the authors oversimplify the problem.

We have access to the distance between cells through the Andrew’s score of each cell. However, the challenge is that these distances are relative values and specific to each dataset. While we believe that these distances could provide valuable information, we have not yet determined the most effective way to represent and utilize this data in a meaningful manner.

Also, the discussion about the pipeline's uniqueness does not go into the details of alternative models.The introduction remains weak in outlining the limitations of current methods (L90). Acknowledging this limitation will be necessary.

Thank you for these insightful comments. The discussion about alternative methods was already present in the discussion L586-598 but to answer the request of the reviewers, we have revised the introduction and discussion sections to more clearly address the limitations of current methods, as well as discussed the uniqueness of the pipeline. Additionally, we have reorganized Figure 1 to more effectively highlight the main caveats associated with clustering, the primary method currently in use.

(2) The manuscript suffers from several overstatements and simplifications, which need to be resolved. For example:a) L40: The authors talk about "accurately ranked cells". Based on their results, the term "accuracy" is still unclear in this context.

Thank you for this comment. Our use of the term "accurately" was intended to convey that the ranking was correct based on comparison with human experts, though we agree that it may have been overstated. We have removed "accurately" and propose to replace it with "properly" to better reflect the intended meaning.

b) L50: Microglial processes are not necessarily evenly distributed in the healthy brain. Depending on their embedded environment, they can have longer process extensions (e.g., frontal cortex versus cerebellum).

Thank you for raising this point to our attention. We removed evenly to be more inclusive on the various morphologies of microglia cells in this introductory sentence

c) L69: The term "metabolic challenge" is very broad, ranging from glycolysis/FAO switches to ATP-mediated morphological adaptations, and it needs further clarification about the author's intended meaning.

Thank you for this comment, indeed we clarified to specify that we were talking about the metabolic challenge triggered by ischemia and added a reference as well.

d) L75: Is morphology truly "easy" to obtain?

Yes, it is in comparison to other parameters such as transcripts or metabolism, but we understand the point made by the reviewer and we found another way of writing it. As an alternative we propose: “morphology is an indicator accessible through…”

e) L80: The sentence structure implies that clustering or artificial intelligence (AI) are parameters, which is incorrect. Furthermore, the authors should clarify the term "AI" in their intended context of morphological analysis.

We apologize for this confusing writing, we reformulated the sentence as follows: “Artificial intelligence (AI) approaches such as machine learning have also been used to categorize morphologies (Leyh et al., 2021)”.

f) L390f: An assumption is made that the contralateral hemisphere is a non-pathological condition. How confident are the authors about this statement? The brain is still exposed to a pathological condition, which does not stop at one brain hemisphere.

We did not say that the contralateral is non-pathological but that the microglial cells have a non-pathological morphology which is slightly different. The contralateral side in ischemic experiments is classically used as a control (Rutkai et al 2022). Although It has been reported that differences in transcript levels can be found between sham operated animals and contralateral hemisphere in tMCAo mice (Filippenkov et al 2022) https://doi.org/10.3390/ijms23137308 showing that indeed the contralateral side is in a different state that sham controls, no report have been made on differences in term of morphology.

We have removed “non-pathological” to avoid misinterpretations

g) Methodological questions:a) L299: An inversion operation was applied to specific parameters. The description needs to clarify the necessity of this since the PCA does not require it.

Indeed, we are sorry for this lack of explanation. Some morphological indexes rank cells from the least to the most ramified, while others rank them in the opposite order. By inverting certain parameters, we can standardize the ranking direction across all parameters, simplifying data interpretation. This clarification has been added to the revised manuscript as follows:

“Lacunarity, roundness factor, convex hull radii ratio, processes cell areas ratio and skeleton processes ratio were subjected to an inversion operation in order to homogenize the parameters before conducting the PCA: indeed, some parameters rank cells from the least to the most ramified, while others rank them in the opposite order. By inverting certain parameters, we can standardize the ranking direction across all parameters, thus simplifying data interpretation.”

b) Different biological samples have been collected across different species (rat, mouse) and disease conditions (stroke, Alzheimer's disease). Sex is a relevant component in microglia morphology. At first glance, information on sex is missing for several of the samples. The authors should always refer to Table 1 in their manuscript to avoid this confusion. Furthermore, how many biological animals have been analyzed? It would be beneficial for the study to compare different sexes and see how accurate Andrew's ranking would be in ranking differences between males and females. If they have a rationale for choosing one sex, this should be explained.

As reported in the literature, we acknowledge the presence of sex differences in microglial cell morphology. Due to ethical considerations and our commitment to reducing animal use, we did not conduct dedicated experiments specifically for developing MorphoCellSorter. Instead, we relied on existing brain sections provided by collaborators, which were already prepared and included tissue from only one sex—either female or male—except in the case of newborn pups, whose sex is not easily determined. Consequently, we were unable to evaluate whether MorphoCellSorter is sensitive enough to detect morphological differences in microglia attributable to sex. Although assessing this aspect is feasible, we are uncertain if it would yield additional insights relevant to MorphoCellSorter’s design and intended applications.

To address this, we have included additional references in Table 1 of the revised manuscript and clearly indicated the sex of the animals from which each dataset was obtained.

c) In the methodology, the slice thickness has been given in a range. Is there a particular reason for this variability?

We could not spot any range in the text, we usually used 30µm thick sections in order to have entire or close to entire microglia cells.

Although the thickness of the sections was identical for all the sections of a given dataset, only the plans containing the cells of interest were selected during the imaging for both of the ischemic stroke model. This explains why depending on how the cell is distributed in Z the range of the plans acquired vary.Also, the slice thickness is inadequate to cover the entire microglia morphology. How do the authors include this limitation of their strategy? Did the authors define a cut-off for incomplete microglia?

We found that 30 µm sections provide an effective balance, capturing entire or nearly entire microglial cells (consistent with what we observe *in vivo*) while allowing sufficient antibody penetration to ensure strong signal quality, even at the section's center. In our segmentation process, we excluded microglia located near the section edges (i.e., cells with processes visible on the first or last plane of image acquisition, as well as those close to the field of view’s boundary). Although our analysis pipeline should also function with thicker sections (>30 µm), we confirmed that thinner sections (15 µm or less) are inadequate for detecting morphological differences, as tested initially on the AD model. Segmented, incomplete microglia lack the necessary structural information to accurately reflect morphological differences thus impairing the detection of existing morphological differences.

c) The manuscript outlines that the authors have used different preprocessing pipelines, which is great for being transparent about this process. Yet, it would be relevant to provide a rationale for the different imaging processing and segmentation pipelines and platform usages (Supplementary Figure 7). For example, it is not clear why the Z maximum projection is performed at the end for the Alzheimer's Disease model, while it's done at the beginning of the others.The same holds through for cropping, filter values, etc. Would it be possible to analyze the images with the same pipelines and compare whether a specific pipeline should be preferable to others?

The pre-processing steps depend on the quality of the images in each dataset. For example, in the AD dataset, images acquired with a wide-field microscope were considerably noisier compared to those obtained via confocal microscopy. In this case, reducing noise plane-by-plane was more effective than applying noise reduction on a Z-projection, as we would typically do for confocal images. Given that accurate segmentation is essential for reliable analysis in MorphoCellSorter, we chose to tailor the segmentation approach for each dataset individually. We recommend future users of MorphoCellSorter take a similar approach. This clarification has been added to the discussion.

On a note, Matlab is not open-access,

This is correct. We are currently translating this Matlab script in Python, this will be available soon on Github. https://github.com/Pascuallab/MorphCellSorter.

This also includes combining the different animals to see which insights could be gained using the proposed pipelines.

Because of what we have been explaining earlier, having a common segmentation process for very diverse types of acquisitions (magnification, resolution and type of images) is not optimal in terms of segmentation and accuracy in the analysis. Although we could feed MorphoCellSorter with all this data from a unique segmentation pipeline, the results might be very difficult to interprete.

d) L227: Performing manual thresholding isn't ideal because it implies the preprocessing could be improved. Additionally, it is important to consider that morphology may vary depending on the thresholding parameters. Comparing different acquisitions that have been binarized using different criteria could introduce biases.

As noted earlier, segmentation is not the main focus of this paper, and we leave it to users to select the segmentation method best suited to their datasets. Although we acknowledge that automated thresholding would be in theory ideal, we were confronted toimage acquisitions that were not uniform, even within the same sample. For instance, in ischemic brain samples, lipofuscin from cell death introduces background noise that can artificially impact threshold levels. We tested global and local algorithms to automatically binarize the cells but these approaches resulted often on imperfect and not optimized segmentation for every cell. In our experience, manually adjusting the threshold provides a more accurate, reliable, and comparable selection of cellular elements, even though it introduces some subjectivity. To ensure consistency in segmentation, we recommend that the same person performs the analysis across all conditions. This clarification has been added to the discussion.

e) Parameter choices: L375: When using k-means clustering, it is good practice to determine the number of clusters (k) using silhouette or elbow scores. Simply selecting a value of k based on its previous usage in the literature is not rigorous, as the optimal number of clusters depends on the specific data structure. If they are seeking a more objective clustering approach, they could also consider employing other unsupervised techniques, (e.g. HDBSCAN) (L403f).

We do agree with the referee’s comment but, the purpose of the k-mean we used was just to illustrate the fact that the clusters generated are artificial and do not correspond to the reality of the continuum of microglia morphology. In the course of the study we used the elbow score to determine the k means but this did not work well because no clear elbow was visible in some datasets (probably because of the continuum of microglia morphologies). Anyway, using whatever k value will not change the problem that those clusters are quite artificial and that the boundaries of those clusters are quite arbitrary whatever the way k is determined manually or mathematically.

L373: A rationale for the choice of the 20 non-dimensional parameters as well as a detailed explanation of their computation such as the skeleton process ratio is missing. Also, how strongly correlated are those parameters, and how might this correlation bias the data outcomes?

Thank you for raising this point. There is no specific rationale beyond our goal of being as exhaustive as possible, incorporating most of the parameters found in the literature, as well as some additional ones that we believed could provide a more thorough description of microglial morphology.

Indeed, some of these parameters are correlated. Initially, we considered this might be problematic, but we quickly found that these correlations essentially act as factors that help assign more weight to certain parameters, reflecting their likely greater importance in a given dataset. Rather than being a limitation, the correlated parameters actually enhance the ranking. We tested removing some of these parameters in earlier versions of MorphoCellSorter, and found that doing so reduced the accuracy of the tool.

Differences between circularity and roundness factors are not coming across and require further clarification.

These are two distinct ways of characterizing morphological complexity, and we borrowed these parameters and kept the name from the existing literature, not necessarily in the context of microglia. In our case, these parameters are used to describe the overall shape of the cell. The advantage of using different metrics to calculate similar parameters is that, depending on the dataset, one method may be better suited to capture specific morphological features of a given dataset. MorphoCellSorter selects the parameter that best explains the greatest dispersion in the data, allowing for a more accurate characterization of the morphology. In Author response image 1 you will see how circularity and roundness describe differently cells

**Author response image 1. sa3fig1:** Correlation between Circularity and Roundness Factor in the Alzheimer disease dataset. A second order polynomial correlation exists between the two parameters in our dataset. Indeed (1) a single maximum is shared between both parameters. However, Circularity and Roundness Factor are not entirely redundant, as examplified by (2) the possible variety of Roundness Factors for a given Circularity as well as (3) the very different morphology minima of these two parameters.

One is applied to the soma and the other to the cell, but why is neither circularity nor loudness factor applied to both?

None of the parameters concern the cell body by itself. The cell body is always relative to another metric(s). Because these parameters and what they represent does not seem to be very clear we have added a graphic representation of the type of measurements and measure they provide in the revised version of the manuscript (Supplemental figure 8).

f) PCA analysis:The authors spend a lot of text to describe the basic principles of PCA. PCA is mathematically well-described and does not require such depth in the description and would be sufficient with references.

Thank you for this comment indeed the description of PCA may be too exhaustive, we will simplify the text.

Furthermore, there are the following points that require attention:L321: PC1 is the most important part of the data could be an incorrect statement because the highest dispersion could be noise, which would not be the most relevant part of the data. Therefore, the term "important" has to be clarified.

We are not sure in the case of segmented images the noise would represent most of the data, as by doing segmentation we also remove most of the noise, but maybe the reviewer is concerned about another type of noise? Nonetheless, we thank the reviewer for his comment and we propose the following change, that should solve this potential issue.

“*PC*_1<.sub> is the direction in which data is most dispersed.”_

L323: As before, it's not given that the first two components hold all the information.

Thank you for this comment we modified this statement as follows: “The two first components represent most of the information (about 70%), hence we can consider the plan *PC*_1_, *PC*_2_ as the principal plan reducing the dataset to a two dimensional space”

L327 and L331 contain mistakes in the nomenclature: Mix up of "wi" should be "wn" because "i" does not refer to anything. The same for "phi i = arctan(yn/wn)" should be "phi n".

Thanks a lot for these comments. We have made the changes in the text as proposed by the reviewer.

L348: Spearman's correlation measures monotonic correlation, not linear correlation. Either the authors used Pearson Correlation for linearity or Spearman correlation for monotonic. This needs to be clarified to avoid misunderstandings.

Sorry for the misunderstanding, we did use Spearman correlation which is monotonic, we thus changed linear by monotonic in the text. Thanks a lot for the careful reading.

g) If the authors find no morphological alteration, how can they ensure that the algorithm is sensitive enough to detect them? When morphologies are similar, it's harder to spot differences. In cases where morphological differences are more apparent, like stroke, classification is more straightforward.

We are not entirely sure we fully understand the reviewer's comment. When data are similar or nearly identical, MorphoCellSorter performs comparably to human experts (see Table 1). However, the advantage of using MorphoCellSorter is that it ranks cells do.much faster while achieving accuracy similar to that of human experts AND gives them a value on an axis (andrews score), which a human expert certainly can't. For example, in the case of mouse embryos, MorphoCellSorter’s ranking was as accurate as that made by human experts. Based on this ranking, the distributions were similar, suggesting that the morphologies are generally consistent across samples.

The algorithm itself does not detect anything—it simply ranks cells according to the provided parameters. Therefore, it is unlikely that sensitivity is an issue; the algorithm ranks the cells based on existing data. The most critical factor in the analysis is the segmentation step, which is not the focus of our paper. However, the more accurate the segmentation, the more distinct the parameters will be if actual differences exist. Thus, sensitivity concerns are more related to the quality of image acquisition or the segmentation process rather than the ranking itself. Once MorphoCellSorter receives the parameters, it ranks the cells accordingly. When cells are very similar, the ranking process becomes more complex, as reflected in the correlation values comparing expert rankings to those from MorphoCellSorter (Table 1).

Moreover, MorphoCellSorter does not only provide a ranking: the morphological indexes automatically computed offer useful information to compare the cells’ morphology between groups.

h) Minor aspects:% notation requires to include (weight/volume) annotation.

This has been done in the revised version of the manuscript

Citation/source of the different mouse lines should be included in the method sections (e.g. L117).

The reference of the mouse line has been added (RRID:IMSR_JAX:005582) to the revised version of the manuscript.

L125: The length of the single housing should be specified to ensure no variability in this context.

The mice were kept 24h00 individually, this is now stated in the text

L673: Typo to the reference to the figure.

This has been corrected, thank you for your thoughtful reading.

**Recommendations for the authors:**

**Reviewer #1 (Recommendations for the authors):**
Methods(1) Alzheimer's disease model: was a perfusion performed and then an hour later brains extracted? Please clarify.

This is indeed what has been done.

(2) For *in vitro* microglial studies: was a percoll gradient used for the separation of immune cells? What percentage percoll was used? Was there separation of myelin and associated debris with the percoll centrifugation? Please clarify the protocol as it is not completely clear how these cells were separated from the initial brain lysate suspension. What cell density was plated?

The protocol has been completed, as followed: “Myelin and debris were then eliminated thanks to a Percoll PLUS solution (E0414, Sigma-Aldrich) diluted with DPBS10X (14200075, Gibco) and enriched in MgCl_2_ and CaCl_2_ (for 50 mL of myelin separation buffer: 90 mL of Percoll PLUS, 10 mL of DPBS10X, 90 μL of 1 M CaCl_2_ solution, and 50 μL of 1 M MgCl_2_ solution).”. Thank you for your feedback.

(3) How are the microglia "automatically cropped" in FIJI (for the Phox2b mutant)? Is there a function/macro in the program you used? This is very important for the workflow and needs to be clarified. The methods section of this manuscript is a guide for future users of this workflow and should be as descriptive as possible. It would be useful to give detailed information on the manual classification process, perhaps as a supplement. The authors do a nice job pointing out that these older methods are not effective in categorizing microglia that don't necessarily fit into a predefined phenotype.

The protocol has been completed, as follows “. Briefly, the centroid of each detected object (i.e. microglia), except the ones on the borders, were detected, and a crop of 300x300 pixels around the objects were generated. Then, the pixels belonging to neighboring cells were manually removed on each generated crop.

(4) Please address the concern that manual tuning and thresholding are required for this method's accuracy. Is this easily reproducible?

Yes, it is easily reproducible for a given experimenter and is better suited than automatic thresholding. Although segmentation is not the primary focus of this paper, we leave it to users to choose the segmentation method that best fits their datasets.

To address your question, we acknowledge that automated thresholding would theoretically be ideal. However, we encountered challenges due to non-uniform image acquisitions, even within the same sample. For instance, in ischemic brain samples, lipofuscin resulting from cell death introduced background noise that could artificially influence threshold levels. We tested both global and local algorithms for automatic binarization of cells, but these approaches often produced suboptimal segmentation results for individual cells.

Based on our experience, manually adjusting the threshold provided more accurate, reliable, and consistent selection of cellular elements, even though it introduces a degree of subjectivity. To maintain consistency, we recommend that the same individual perform the analysis across all conditions.

This clarification has been incorporated into the discussion as follows: “Although, automated thresholding would be ideal. In our case, image acquisitions were not entirely uniform, even within the same sample. For instance, in ischemic brain samples, lipofuscin from cell death introduces background noise that can artificially impact threshold levels. This effect is observed even when comparing contralateral and ipsilateral sides of the same brain. In our experience, manually adjusting the threshold provides a more accurate, reliable, and comparable selection of cellular elements, even though it introduces some subjectivity. To ensure consistency in segmentation, we recommend that the same person performs the analysis across all conditions. “

(5) How are the authors performing the PCA---what program (e.g .R)? Again, please be explicit about how these mathematical operations were computed. (lines 302-345).

The PCA was made in Matlab, the code can be found on Github (https://github.com/Pascuallab/MorphCellSorter), as stated in the discussion.

Other:(1) Can the authors comment on the challenges of the *in vitro* microglial analyses? The correlation of the experts v. MorphoCellSorter is much less than the fixed tissue. This is not addressed in the manuscript.

*In vitro*, microglial cells exhibit a narrower range of morphological diversity compared to *ex vivo* or *in vivo* conditions. A higher proportion of cells share similar morphologies or morphologies with comparable complexities, which makes establishing a precise ranking more challenging. Consequently, the rank of many cells could be adjusted without significantly affecting the overall quality of the ranking.

This explains why the rankings tend to show slightly greater divergence between experts. Interestingly, the ranking generated by MorphoCellSorter, which is objective and not subject to human bias, lies roughly midway between the rankings of the two experts.

(2) You point out that the MorphoCellSorter may not be suited for embryonic/prenatal microglial analysis.

This must be a misunderstanding because it is not what we concluded; we found that the ranking was correct but that we could not spot any differences due to transgenic alteration.

The lack of differences observed in the embryonic microglia (Figure 5) is not necessarily surprising, as embryonic microglia have diverse morphological characteristics--- immature microglia do not possess highly ramified processes until postnatal development [see Hirosawa et al. (2005) https://doi.org/10.1002/jnr.20480 -they use an Iba1-GFP transgenic mouse to visualize prenatal microglia]. Also, see Bennett et al. (2016) [https://doi.org/10.1073/pnas.1525528113] which shows mature microglia not appearing until 14 days postnatal.

We agree with the reviewer on that point nonetheless MorphoCellSorter provides an information on the fact that the population is homogeneous and that the mutation has no effect on the morphology.

(3) Although a semantic issue, Figure 1's categorization of microglia shows predefined groups of microglia do not necessarily usefully bin many cells. Is still possible to categorize the microglia without using hotly debated categorization methods? The literature review in the current manuscript correctly points out the spectrum phenomenon of microglial activation states, though some of the suggestions from Paolicelli et al. (2022) are not put into action. The use of "activated" only further perpetuates the oversimplified classification of microglia. Perhaps the authors could consider using the term "reactive", as it is recognized by the Microglial nomenclature paper cited above. Are "amoeboid microglia" not "activated microglia"? "Reactive" is a less loaded term and is a recommended descriptor. Amoeboid microglia are commonly understood to be indicative of a highly proinflammatory environment, though you could potentially use "hyper-reactive" to differentiate them from the slightly ramified "reactive" cells.

We changed activated microglia to reactive microglia as requested by the reviewer in the text. Thanks a lot for your comment

(4) The graphs in Figures 3 B-D are visually difficult to interpret. The better color contrast between the MorphoCellSorter/Expert and Expert1/Expert2 would be useful--- perhaps a color for Expert 1 and a different color for Expert 2. Is this the ranking from the same data in Figure 1 (lines 420-421)? It is unclear what the x-axis represents in 3B-D. E-G is much more intuitive.

We believe the confusion stems more from Figure 1 than Figure 3, as both figures use similar representations for entirely different analyses (clustering vs. ranking). To address this, we have provided an updated version of Figure 1 to help clarify this distinction and avoid any potential misinterpretation.

Regarding Figure 3B-D, we do not fully see the need for changing the colors. These panels are histograms that display the distribution of rank differences either between experts and MorphoCellSorter or between the two experts. Assigning specific colors to the experts or MorphoCellSorter would be challenging, as the histograms represent comparative distributions involving both an expert and MorphoCellSorter or the ranking differences between the two experts.

The same reasoning applies to Figures 3E-G. In these scatter plots, each point is defined by an ordinate (ranking value for one expert) and an abscissa (ranking value for either the other expert or MorphoCellSorter). Therefore, it would not be straightforward or meaningful to assign distinct colors to these elements within this context.

(5) Line 217: use the term "imaged" rather than "generated" ... or "images were generated of clusters of microglia located .... using MICROSOPE and Zen software." You aren't generating microglia, rather, you are generating images.

Thanks a lot for raising this problem, we changed the sentence as followed: “For the AD model, crops of individual microglial cells located in the secondary visual cortex were extracted from images using the *Zen* software (v3.5, Zeiss) and exported to the Tif image format.

(6) Elaborate on how an "inversion operation" was applied to Lacunarity, roundness factor, convex hull radii ratio, processes cell areas ratio, and skeleton processes. (Lines 299-300) Furthermore, a paragraph separation would be useful if the "inversion operation" is not what is described in the text immediately after this description.

Indeed, we are sorry for this lack of explanation. Some morphological indexes rank cells from the least to the most ramified, while others rank them in the opposite order. By inverting certain parameters, we can standardize the ranking direction across all parameters, simplifying data interpretation. This clarification has been added to the revised manuscript as follows:

“Lacunarity, roundness factor, convex hull radii ratio, processes cell areas ratio and skeleton processes ratio were subjected to an inversion operation in order to homogenize the parameters before conducting the PCA: indeed, some parameters rank cells from the least to the most ramified, while others rank them in the opposite order. By inverting certain parameters, we can standardize the ranking direction across all parameters, thus simplifying data interpretation.”

(7) Line 560: "measureclarke" seems to be an error associated with the reference. Please correct.

Thanks a lot, this has been corrected

(8) Discussion: compare MorphoCellSorter to the MIC-MAC program used by Salamanca et al. (2019). They use a similar approach, albeit not Andrew's plot.

We have added the Salamanca reference

**Reviewer #2 (Recommendations for the authors):**
While it's not expected that the authors address the significance of the morphology in relation to function here, they could help highlight the issue and produce data that would enhance the paper's significance. Therefore, I recommend a small-scale and straightforward study where the authors couple their analysis with a marker (e.g. Lysotracker or Mitotracker) to produce data that link their morphometric analysis to more functional readouts. Furthermore, I encourage the authors to elaborate on the practical applications of these morphometric tools and the implications of their measurements, as this would provide context for their work, which, as it stands, feels like just another tool.

We would like to thank the reviewer for their thoughtful comment and suggestion. Indeed, MorphoCellSorter is simply another tool, but one that offers a more convenient and efficient approach, producing a variety of results tailored to specific research needs. We strongly believe that MorphoCellSorter should be used in conjunction with other tools, depending on the specific research question.

In our view, MorphoCellSorter is particularly well-suited for researchers who need a quick and efficient way to determine whether their treatment, gene invalidation, or other experimental conditions affect microglial morphology. In this context, MorphoCellSorter is fast, user-friendly, and highly effective. However, for those who aim to uncover detailed differences in cell morphology, other tools requiring more time-intensive, full reconstructions of the cells would be more appropriate.

Providing additional data on the relationship between cellular function and morphology could certainly pave the way for new questions and more robust evidence. For instance, combining single-cell transcriptomics with morphological analysis would be an excellent approach to exploring the relationship between function and morphology. However, this would involve significant time, expense, and effort, and it represents a different line of inquiry altogether.

While it would be ideal to clearly demonstrate the link between morphology and function, we are concerned that pursuing such a goal would considerably delay the implementation and adoption of our tool, potentially raising additional questions beyond the scope of this study.!

Minor comments:(1) Can MorphCellSorter be adapted for use with other cell types (e.g., astrocytes)?

Yes it could, we have made some pretty conclusive analysis on astrocytes but some parameters have to be adapted before being released.

(2) What modifications would be necessary? If it is not applicable, would a name that includes "Microglia" be more descriptive?

Modification would be quite minor, it is mainly the parameters being considered that would change, this is the reason why we will keep the MorphoCellSorter name. Thank you for the suggestion!

(3) A common challenge with such tools is the technical expertise required to use them. Could a user-friendly interface be developed to better fulfill its intended purpose and benefit the community?

This is a good point thank you, and the answer is yes, we will translate our Matlab code to Python to open it to a wider audience and we will certainly work on a friendly user interface!

(4) Given that this tool relies on imaging, can users trace a cell (or group of cells) back to the original image?

Yes, it is possible if each crop is annotated with the spatial coordinates during the segmentation step. It is not yet implemented in the actual version of the software but mainly depend on the way segmentation is performed, which is not the topic of the paper.

(5) Line 36: The "biologically relevant" statement is central and needs to be expanded.

This is not easy as it is the abstract with a word limit. What we mean by this sentence is that when classifying cells we force them by mathematical tools to enter in a group of cells based on metrics that have not necessarily a biological meaning. We suggest the following modification “However, this classification may lack biological relevance, as microglial morphologies represent a continuum rather than distinct, separate groups, and do not correspond to mathematically defined, clusters irrelevant of microglial cells function.”

(6) Line 49-50: Provide reference and elaborate. For example, does this apply during early life?

We have slightly changed the sentence and added a reference.

(7) Line 69: Provide reference.

The reference, Hubert et al 2021 has been added

(8) Lines 78-88: A table summarizing other efforts in morphometric characterization of microglia would be helpful in distinguishing your work from others.

This has already been done in some review articles; we thus added the references to address readers to these reviews. Here is the revised version of the sentence: “ To date, the literature contains a wide variety of criteria to quantitatively describe microglial morphology, ranging from descriptive measures such as cell body surface area, perimeter, and process length to indices calculating different parameters such as circularity, roundness, branching index, and clustering (Adaikkan et al., 2019; Heindl et al., 2018; Kongsui, Beynon, Johnson, & Walker, 2014; Morrison et al., 2017; Young & Morrison, 2018)”

(9) Lines 130, 145: Please provide complete genotype information and the sources of the animals used.

It has been done

(10) Materials and Methods:(1) Standardize the presentation of products (e.g., using # consistently).

It has been done

(2) Provide versions of software used.

We have modified accordingly

(3) Lines 372-373: A table listing the 20 parameters with brief explanations (as partially done in Materials and Methods) would greatly improve readability.

This is done in supp figure 8

(4) Since nomenclature is a critical issue in the literature, you used specific definitions (lines 376-383). However, please indicate (with a reference) why you use the term "activated," as it implies that the others are non-activated. Alternatively, define "activated" cluster differently.

We change activated microglia to reactive microglia as requested by the reviewer #1.

(4) Figure 1: In my opinion placing this figure as the first main figure is problematic as it confuses the message of the paper. Since the authors are introducing a new approach for morphological characterization in Figure 2, I recommend the latter for the sake of readability and clarity should be the first main image, while Figure 1 can move the supplements.

We do agree with the reviewer, we thus changed figure one as explained earlier to reviewer 1. Nonetheless because it is an important step of our reflection process we believe it can stay as a figure. We hope the change made in figure one clarifies the message of the paper.

(5) Figure 1: Please indicate on the figure the marker for the analysis.

Figure 2 has been changed

(6) No funding agencies are communicated.

This has been corrected